# Spatial repellents transfluthrin and metofluthrin affect the behavior of *Dermacentor variabilis*, *Amblyomma americanum*, and *Ixodes scapularis* in an *in vitro* vertical climb assay

Eric L. Siegel[1], Marcos Olivera[2], Esteban Martinez Roig [2], Melynda Perry[3], Andrew Y. Li[4], Sebastián D'hers [2], Noel M. Elman[5], Stephen M. Rich [1] *

**1** Department of Microbiology, Laboratory of Medical Zoology, University of Massachusetts, Amherst, Massachusetts, United States of America, **2** Computational Mechanics Center, Instituto Tecnológico de Buenos Aires (ITBA), Ciudad Autónoma de Buenos Aires, Argentina, **3** Textile Materials Evaluation Team, Combat Capabilities Development Command Soldier Center, United States Army Garrison-Natick, Natick, Massachusetts, United States of America, **4** United States Department of Agriculture, Agricultural Research Service, Invasive Insect Biocontrol & Behavior Laboratory, Beltsville, Maryland, United States of America, **5** GearJump Technologies, Limited Liability Company, Brookline, Massachusetts, United States of America

* smrich@umass.edu

## Abstract

Repellents serve an important role in bite protection. Tick repellents largely rely on biomechanisms that induce responses with direct contact, but synthetic pyrethroids used as spatial repellents against insects have received recent attention for potential use in tick protection systems. An *in vitro* vertical climb assay was designed to assess spatial repellency against *Dermacentor variabilis*, *Amblyomma americanum*, and *Ixodes scapularis* adult, female ticks. Climbing behavior was assessed with and without the presence of two spatial repellents, transfluthrin and metofluthrin. Repellency parameters were defined to simulate the natural questing behavior of ambushing ticks, including measures of detachment, pseudo-questing duration, climbing deterrence, and activity. Significant effects were observed within each parameter. *D. variabilis* showed the greatest general susceptibility to each repellent, followed by *A. americanum*, and *I. scapularis*. The most important and integrative measure of repellency was climbing deterrence–a measure of the spatial repellent's ability to disrupt a tick's natural propensity to climb. Transfluthrin deterred 75% of *D. variabilis*, 67% of *A. americanum*, and 50% of *I. scapularis*. Metofluthrin was slightly more effective, deterring 81% of *D. variabilis*, 73% of *A. americanum*, and 72% of *I. scapularis*. The present study poses a novel paradigm for repellency and reports a preliminary assessment of spatial repellent effect on tick behavior. Further research will assess spatial repellency in a more natural setting, scale exposure conditions, and incorporate host cues.

**Data Availability Statement:** All relevant data are within the paper and its Supporting Information files.

**Funding:** This study was supported by the DoD Tick-borne Disease Program from the Congressional Directed Medical Research Program, grant # W81XWH1920028. The funders had no role in study design, data collection and analysis, decision to publish, or preparation of the manuscript.

**Competing interests:** The authors have declared that no competing interests exist.

# 1. Introduction

Ticks are the principal arthropod vectors of many prevalent zoonotic diseases in North America, including Lyme disease, anaplasmosis, and babesiosis [1, 2]. The incidence of tick-borne disease, though vastly under-reported, has risen dramatically in the past decade. Estimates in the United States illustrate this trend, with yearly incident case counts growing from 20,000 (2004) to 50,000 (present) [3, 4]. This rise is due to a combination of entomological factors that promote natural disease spread (including increases in tick geographical range/population density and shifts in host population dynamics) and human behaviors that constitute risk factors (such as pet ownership, outdoor activity, and the absence or misuse of personal protective measures) [5–7]. Addressing the burden of tick-borne disease therefore requires interdisciplinary approaches that target anthropogenic and entomological contributions to disease transmission [8].

Tick bite defense relies on personal protection measures [9]. They include simple, cost-effective strategies that at-risk individuals can use to limit exposure or contact time with ticks. Tactics may involve performing tick checks, running clothing worn outdoors in a hot dryer, and repellent or acaricide use [10, 11]. The use of repellents is an efficacious and affordable option for protection [12]. In many cases, repellents offer protection from a range of disease-causing, hematophagous arthropods, making them especially useful in regions of overlapping vector presence [13–15]. DEET (*N, N-diethyl-3-methylbenzamide)* is the most prominent commercial arthropod repellent and the gold standard against which novel formulations are compared [16]. It deters mosquitoes from landing on a treated surface by interfering with odorant receptors on antennae that respond to host cues [17, 18]. In ticks, however, DEET has been described to work as a contact irritant–with varied efficacy across species [19, 20]. Another commonly used repellent is permethrin, a synthetic pyrethroid that kills and repels mosquitoes with direct droplet contact [21]. It is also used as a clothing treatment for long-lasting tick protection through acaricidal action, with little signs of true repellent function [22].

"Repellency" warrants clarification, as the aforementioned examples show that products with varying biomechanisms (such as killing, attraction inhibition, and irritation) are often collectively referred to as "repellents" [23]. Mosquitoes and other insects are fast-moving and spend mere seconds on a host [24]. Repellency efforts that target mosquitoes are consequently focused on actions that prevent landing on a host [25, 26]. Ticks in comparison are relatively slow moving, do not fly, and may spend days attached to a host. The strict sense definition of repellency–wherein the arthropod makes an oriented movement away from the repellent source upon chemoreception–is therefore not always applicable with ticks [27]. A more appropriate definition of repellency in ticks should encompass all modifiable behaviors that may prevent disease transmission throughout the tick-borne disease risk pathway. These include the actions that prevent moving onto a host, locating a favorable site of feeding, and inserting mouthparts for feeding/disease transmission [23, 28].

Repellents can be further classified by their mode of action as contact and spatial repellents depending on whether a target must contact a treated surface or react by non-tactile means to the volatilized state [29–31]. Active ingredients derived from synthetic pyrethroids are frequently used as spatial repellents against vectors that cover large distances, such as mosquitoes and biting flies [32, 33]. The efficacy of these spatial repellents is dependent on inherent arthropod sensitivity and the ability to maintain a consistent vapor phase concentration, which is often managed with controlled release devices (CRDs) that modulate dispersion [34–36]. Tick research has largely focused on contact repellency, leading to the development of guidelines for the quantitative evaluation of contact repellency, with clear standards guiding measurement and interpretation [37]. Recent investigations, however, have explored the

ability of spatial repellents to affect tick behavior [38–42]. Two synthetic pyrethroids, transfluthrin and metofluthrin, have demonstrated the ability to affect tick behavior through repellent and acaricidal properties in the vapor phase [40–42]. Information gained from these studies has led to the need for further evaluation of spatial repellent efficacy and the value these products may hold in personal and environmental protection systems. The present study sought to contribute to the understanding of spatial repellency in ticks by (1) defining an *in vitro* vertical climb assay with specific output parameters that quantify climbing behavior in ticks; (2) evaluating the ability of two CRD-delivered pyrethroids, transfluthrin and metofluthrin, to affect the climbing behavior of *Dermacentor variabilis*, *Amblyomma americanum*, and *Ixodes scapularis* based on these parameters; (3) making recommendations for future work for spatial repellency evaluation in ticks.

## 2. Materials and methods

### 2.1 Spatial repellents

Two spatial repellents were tested: metofluthrin (in-house, 97.15% technical grade: 91.96% Z-Isomer, 5.19% E-Isomer) and transfluthrin (98.68% technical grade, Bayer Corporation, Pittsburgh, PA, USA). Isopropanol (absolute, Sigma-Aldrich, St. Louis, MO, USA) was used as the solvent in preparation of 30% (w/v) formulations of each spatial repellent.

### 2.2 Ticks

**2.2.1 Sourcing.** 162 pathogen-free adult, female ticks: (60 *D. variabilis*, 48 *A. americanum*, and 54 *I. scapularis*) were used and originated from the tick-rearing facility at the Oklahoma State University, Department of Entomology and Plant Pathology, National Tick Research and Educational Resource.

**2.2.2 Storage and handling.** Ticks were stored in 48 mm (h) x 20 mm (w) Fisherbrand polystyrene containers with plastic snap caps. A 5 mm diameter orifice was punched into each top. This hole was covered with woven, nylon mesh (Voile White Decorator Fabric) to prevent ticks from escaping. Ticks were stored four to a vial, and a chilled coma was maintained at 4˚C until the ticks were used. The containers were removed from refrigeration weekly and opened for 10 minutes. The cloth, vials, and containers were checked thoroughly at this time for evidence of fungal growth. If evidence of growth was noted on the vials, the ticks were moved to clean vials. Ticks were handled with autoclaved forceps and paintbrushes. Two hours preceding the repellency trial, ticks were removed from refrigeration and equilibrated in an incubator (Percival 1-30VK environmental chamber) at 23˚C at 90% relative humidity.

### 2.3 Experimental setup

**2.3.1 The controlled release device.** The CRD consisted of a 4.5 cm (l) x 1.5 cm (w) x 2.0 cm (z) 3D printed frame with three porous reservoirs (Fig 1). The three reservoirs (1 cm diameter, 1.5 cm depth) each contained one piece of cellulose sponge (0.5 cm (l) x 0.5 cm (w) x 0.1 cm (z)). The repellent was released through the 1 mm x 1mm square pore on the top face of the device frame.

**2.3.2 Experiment chamber.** The tick behavior test system consisted of (1) a CRD, specifically designed for the sustained spatial release of test repellent formulations (Fig 1), (2) a tick behavioral test chamber (Fig 2), and (3) a computer-based movement tracking system (Ethovision XT, Version 15.0, Noldus Information Technology, Leesburg, VA, USA). The tick behavior test chamber was assembled from six clear, acrylic sheets. Four 60 cm (l) x 30 cm (w) x 0.5 cm (z) sheets formed the bottom, top, front, and back faces, and two 30 cm (l) x 30 cm (w) x

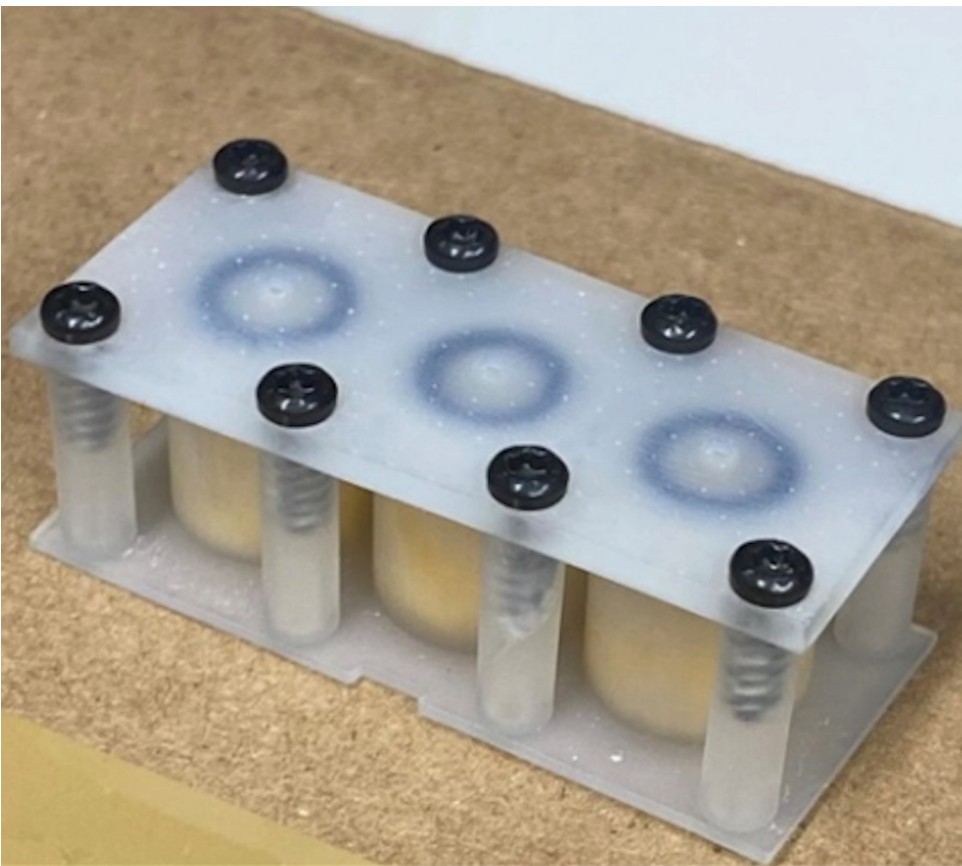

**Fig 1. The controlled release device (CRD).**

0.5 cm (z) sheets formed the sides. Sheets were connected by 90-degree angled plastic hinges, creating a static air chamber. The hinge connectors created a 0.5 cm space around the perimeter of the box, allowing for ventilation.

Three sticks (Simoutal brand polyester-cotton fiber diffuser sticks), 30.5 cm (l) x 0.3 cm (diameter), were used for each climbing experiment. The sticks were adhered to the inside of the top of the chamber by a 1 cm x 1 cm square piece of Crayola air-dry clay, cut with a sterile #10 scalpel. There was a 1.5 cm space between the bottom of the stick and the bottom of the chamber. For a tick to reach the bottom surface of the chamber, it would therefore have to detach and fall from its stick. The sticks were arrayed along the center width of the lid, one centered and the other two 2.5 cm from either side. After each trial, the climbing sticks and clay were discarded, and the walls of the chamber were cleaned with isopropanol (70%).

**2.3.3 Repellency vertical climb bioassay.** The 30% formulation of transfluthrin or metofluthrin (50 µl) was loaded into the middle reservoir of the CRD. The CRD was then adhered to the upper end of the chamber. The repellent was allowed 20 minutes to release into the box from the CRD prior to the introduction of the ticks for the trial (hereafter "induction time"). Trials included 3 female ticks of the same species: one tick per climbing stick.

Ticks were evaluated for inclusion by briefly placing them at the base of an untreated stick. If the tick climbed the stick, it was included in the trial. Ticks were excluded in the rare event that ticks were unable to hold on to the stick, detached, or were unwilling to climb. Climbing trials were conducted by removing the chamber lid with the attached climbing sticks, inverting

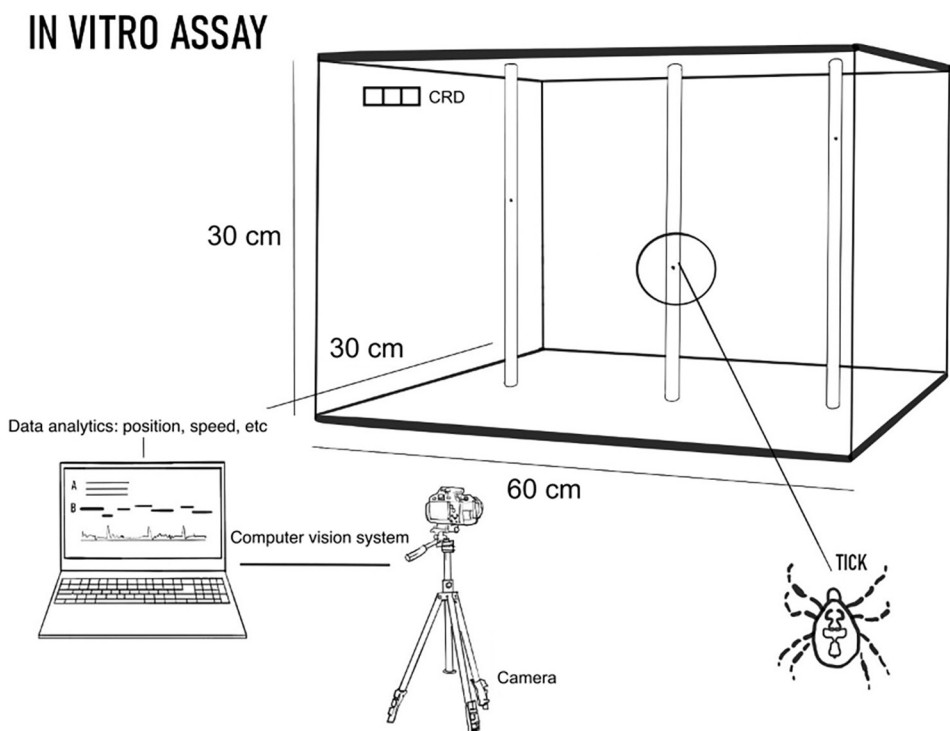

**Fig 2. Experimental setup.** The controlled release device was placed in the upper left-hand corner of the experiment chamber. Three vertical climbing sticks were arrayed along the top face. A camera recorded tick climbing from the perspective shown for quantitative behavior analysis based on vertical movement.

it, and placing the ticks on the climbing sticks. The entire lid with three climbing sticks with ticks at top was then inverted and placed on the walls of the chamber so that the ticks started the trial at the bottom. Tick behavior was observed for 10 minutes. Treatments were compared to controls performed in the same fashion in the absence of repellents.

## 2.4 Video tracking of tick movements

Tick mobility was tracked with a computer vision system [43]. This included a camera (Basler acA1300-60gm) connected to a desktop (Dell Precision 3630) running EthoVision XT. Each experiment was generated with a pre-defined template, maintaining a constant capture rate (3.5 frames per sec.), camera field of view, and detection criteria for tracking. Height was recorded frame by frame during the trial. Minor inaccuracies in tick movement recorded by the tracking system were corrected prior to analysis.

## 2.5 Activity parameters and measures of spatial repellency

Four parameters were defined to measure climbing behavior. These included an analysis of movement and velocity, climbing deterrence, pseudo-questing, and detachment. Each measure was designed to simulate a component of vertical climbing associated with the host-seeking process.

**2.5.1 Activity analysis.** All ticks received a binary score based on if they moved at all during the trial. Ticks that remained stationary at the bottom of their stick during the entire trial were scored "immobile" and ticks that moved at all during the trial were scored "mobile."

Ticks scored "mobile" were considered in a further analysis to see if exposure to the spatial repellents resulted in changes to movement speed. The mean velocity was calculated for each tick from digested height data over the duration of the experiment. Only time points where the tick was moving were included in the analysis. Transfluthrin and metofluthrin groups were compared to controls using the Wilcoxon Rank-Sum Test (WRST) to obtain P values [44]. Pearson's r was used as a measure of effect size to quantify the magnitude of significant associations [45].

**2.5.2 Climbing deterrence.** Ticks were individually scored "not deterred" or "deterred" according to their ability to successfully climb their stick. A tick that was scored "not deterred" was successful in climbing. This required the tick to (1) reach the top of the stick at some point in the trial, *and* (2) not detach from the stick at any point in the trial, *and* (3) have a meaningful presence at the top, indicated by the tick staying at the top of the stick or climbing onto the top surface of the box. Ticks were scored "deterred" if they (1) did not reach the top of the stick by the end of the trial, *or* (2) fell off the stick at any point in the trial, *or* (3) once reaching the top of the stick, moved down. The total "deterred" and "not deterred" in each group were added and presented as proportions. P values were calculated from the difference in proportions between controls and repellent groups with a two-tailed Fisher's exact test. Supplementary effect size calculations were done using Cramer's phi (φ) [46].

**2.5.3 Detachment.** Ticks were scored "detached" if they fell from their stick at any point in the trial. Ticks were scored "not detached" if they did not fall from their stick. Fisher's exact tests and φ values were calculated to evaluate the degree of difference in detachment between controls and repellent groups.

**2.5.4 Pseudo-questing duration.** A pseudo-questing measure was considered to simulate the questing behavior of ambushing ticks. The cumulative duration (in sec.) that each tick was found above the 27 cm mark on their stick was calculated and considered analogous to the time that ticks would spend atop tall grass or other surfaces in the host-seeking process. Time that ticks spent on the lid was included in this measure. The WRST and Pearson's r were used to evaluate the difference between controls and repellent groups.

## 2.6 Data analysis

Analyses were performed in SPSS for Windows, Advanced Statistics, Version 28.0 [47]. Results of WRSTs were considered significant in cases where $W < W_{crit}$ at a significance threshold $\alpha = 0.05$, following SAMPL guidelines [44, 48]. Cohen's classification of effect size (for r and φ) at 1 degree of freedom (0.1–0.3 small, 0.3–0.5 medium, and $> 0.5$ large) provides a standardized interpretation of effect [49, 50]. However, the application to biological investigations varies from system to system [51]. Considering the protective effect expected of a trusted repellent, effect sizes greater than or equal to 0.80 were considered large and of substantial biological significance. Medium effect sizes were considered for values between 0.60 and 0.80. Small effect sizes were considered for values less than 0.60. Output graphs were produced in GraphPad Prism, Version 9.4.1 for macOS [52].

## 2.7 GCMS and *in silico* simulation

Transfluthrin and metofluthrin release rates from the CRD were characterized using analytical formulations, assessing volatilization for 30 minutes, equal to the experiment duration (20-minute induction time and 10-minute trial). For this measurement, the CRD was placed in a sealed plastic bottle and a 50 mL air sample was extracted via syringe chemistry tools based on Gas Chromatography-Mass Spectroscopy Analysis (GCMS). Extracted air samples were dissolved in 5 mL of isopropanol (absolute) and injected directly into the GCMS for the

measurement of chemical concentrations relative to a known standard. Numerical simulations based on computational fluid dynamics (CFD) were also performed. These addressed the transport of evaporated transfluthrin and metofluthrin into still air inside the test chamber. Fluid natural convection, chemical diffusion, and gravity momentum were considered. Detailed information and mathematics on the CFD simulations can be found in supplemental information (S1 and S2 Figs).

## 3. Results

### 3.1 GCMS and CFD simulation

Release rates obtained from the GCMS experiments were 3.1 mg/hr for transfluthrin and 4.4 mg/hr for metofluthrin. Concentration plots from the CFD simulation for both formulations were plotted on a logarithmic scale for a snapshot in time, 25 minutes post-release (Fig 3). Exposure concentrations along the length of each stick were extrapolated from these measurements (Fig 4). Though a slight horizontal concentration gradient was observed, the magnitude

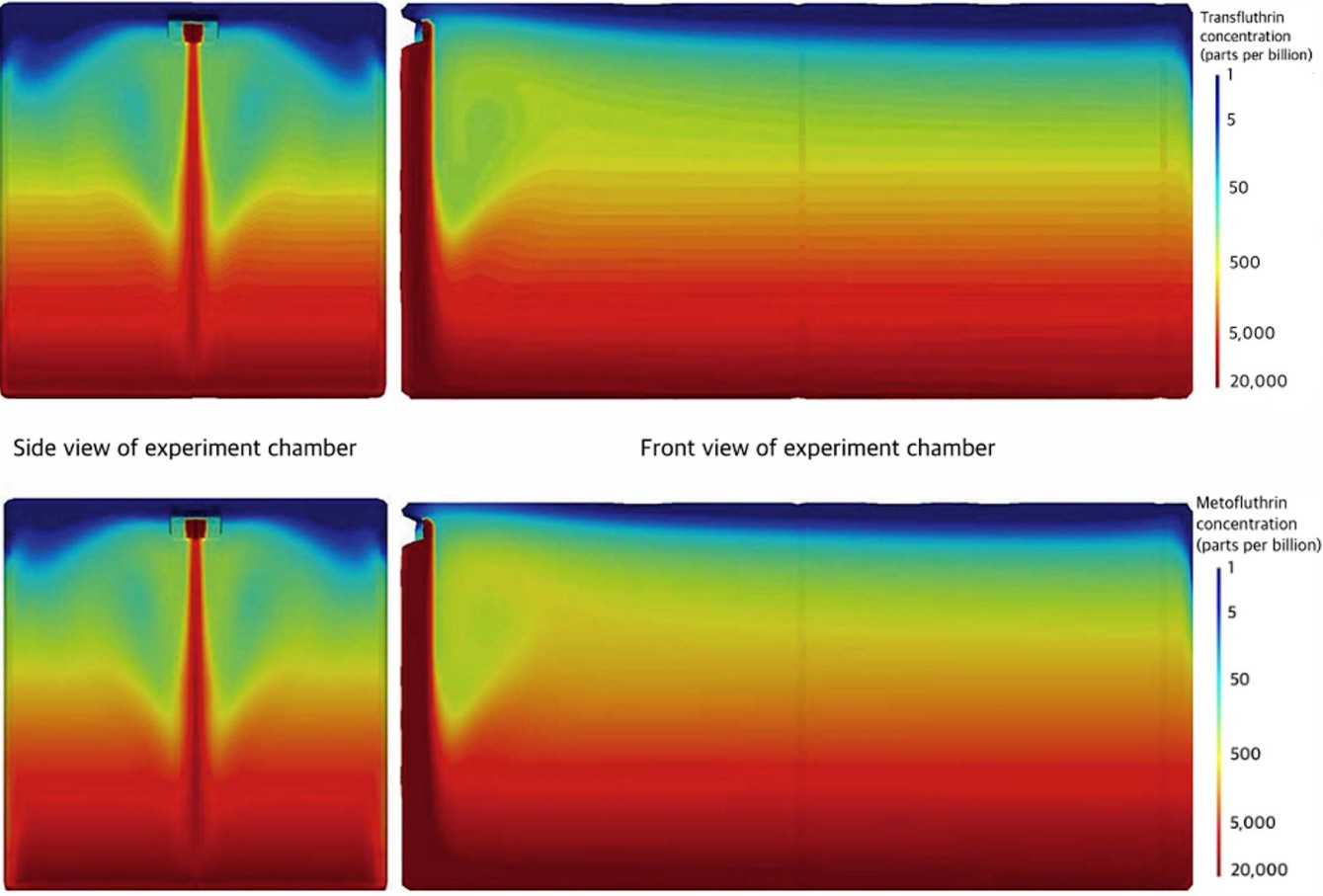

**Fig 3. Concentration gradients from CFD simulation, 25 minutes post-release.** The concentration gradients produced from the CFD simulation were plotted for (A) transfluthrin, and (B) metofluthrin, 25 minutes post-release. Accounting for the 20-minute induction time prior to tick introduction, the concentrations represented tick exposure halfway (5 minutes) through the trial. A vertical concentration gradient was seen, with higher concentrations (warmer colors) present on the bottom of the chamber and lower concentrations (cooler colors) at the top. Two perspectives were illustrated: looking at the chamber from the perspective of the camera (right) and the side with the CRD (left). The CRD was positioned in the top left corner from the perspective of the right box and the upper middle of the side perspective.

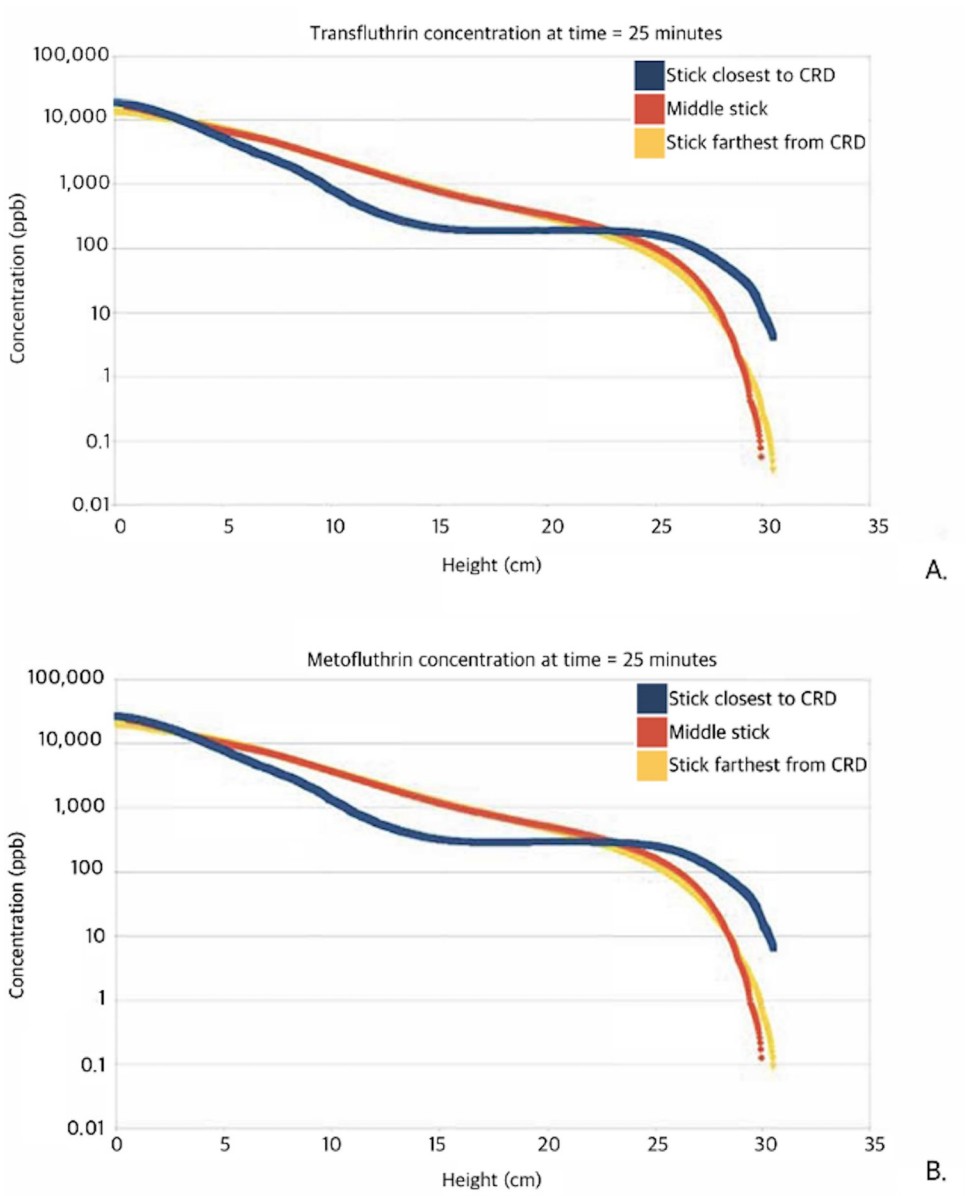

**Fig 4. Exposure profiles of each stick for each repellent, 25 minutes post-release.** Stick position was correlated with the gradients produced in the CFD simulations to show the exposure concentrations on the y-axis (in ppb) with matching position (in cm) along the length of each stick. This was taken 25 minutes post-release (halfway through the 10-minute trials following a 20-minute induction time) for (A) transfluthrin; and (B) metofluthrin.

of the difference in exposure between the three sticks was not noteworthy. Higher concentrations were shown to move precipitously to the bottom of the chamber due to a dispersion flow dominated by gravity. Further, the vapor density of the formulation was greater than that of the still air in the chamber, and this helped push the flow downward immediately from the CRD. Over time, however, the concentration diffused from the bottom of the box to the top, resulting in the formation of a concentration gradient. This gradient persisted over time due to the influence of gravity and the low diffusion rate.

### 3.2 Repellency parameters

**3.2.1 Activity analysis.** All ticks survived the experiments and a period of post-study observation. There was however a noticeable tendency of ticks of each species exposed to transfluthrin to retain a drunken-like state once being removed from the chamber. Ticks presented this behavior by randomly flipping onto their dorsum and showed an inability to walk and climb normally. This effect was not seen in metofluthrin-exposed or control ticks.

Consistent with pre-trial inclusion criteria ensuring that ticks climbed when placed at the base of an untreated stick, all ticks in the control groups for each species were mobile for at least a portion of their trial, readily climbing the stick when placed in the chamber. Each treatment group in contrast showed varying proportions of ticks that remained immobile at the stick bottom for the entirety of the trial, despite showing a willingness to climb just moments prior. This was more prevalent in metofluthrin- exposed ticks than transfluthrin-exposed in each species. 4/21 (19%) of *D. variabilis* exposed to metofluthrin and 3/24 (12.5%) exposed to transfluthrin remained immobile. 4/15 (27%) of *A. americanum* exposed to metofluthrin and 1/18 (6%) exposed to transfluthrin remained immobile. 2/18 (11%) of *I. scapularis* exposed to metofluthrin and 1/18 (6%) exposed to transfluthrin remained immobile. These proportions were not large enough to show significance, however they dictated the sample that was used in the analysis of velocity while moving.

The velocity analysis included the remaining ticks that were scored mobile. Results showed that exposures to both transfluthrin and metofluthrin were associated with significant reductions in tick velocity while moving for *D. variabilis* ($Mdn_{control}$ = 0.365 cm/s; $Mdn_{meto}$ = 0.111 cm/s: P < 0.001, r = 0.79; $Mdn_{trans}$ = 0.110 cm/s: P < 0.001, r = 0.82) and *A. americanum* ($Mdn_{control}$ = 1.041 cm/s; $Mdn_{meto}$ = 0.377: P < 0.001, r = 0.66; $Mdn_{trans}$ = 0.445: P < 0.001, r = 0.71) (Fig 5). The large effect observed with *D. variabilis* also greatly reduced the spread of velocity from the otherwise dispersed control (IQR = 0.185 cm/s), condensing the group

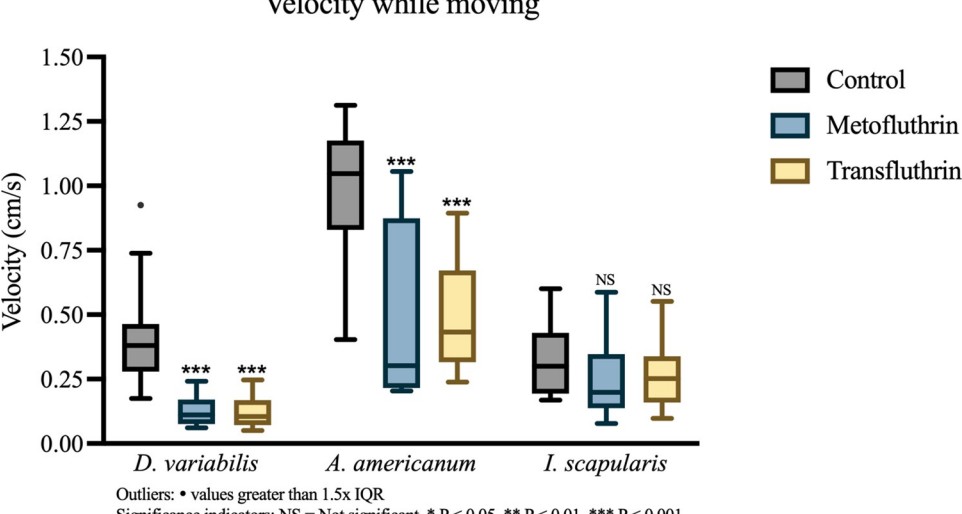

Outliers: • values greater than 1.5x IQR
Significance indicators: NS = Not significant, * P < 0.05, ** P < 0.01, *** P < 0.001

**Fig 5. Comparison of velocity while moving.** Boxplots were made to describe the distribution of tick velocity (in cm/s) while moving along the stick. Points greater than 1.5 x IQR in each group were considered outliers but included in the WRST analysis. Ticks that did not move were excluded. Groups included: N = 147 (*D. variabilis*: 15 control, 17 metofluthrin, 21 transfluthrin; *A. americanum*: 15 control, 11 metofluthrin, 17 transfluthrin; *I. scapularis*: 18 control, 16 metofluthrin, 17 transfluthrin). Significant reductions were shown with transfluthrin and metofluthrin in *D. variabilis* and *A. americanum*. Outliers: • values greater than 1.5x IQR. Significance indicator: NS = Not significant, * P < 0.05, ** P < 0.01, *** P < 0.001.

around a lower median ($IQR_{meto}$ = 0.095 cm/s, $IQR_{trans}$ = 0.098). The opposite effect was observed with *A. americanum* in the metofluthrin group, where the sample, though having a lower median, had an IQR almost twice that of controls ($IQR_{meto}$ = 0.658 cm/s, $IQR_{control}$ = 0.347 cm/s). Transfluthrin exposure in comparison did not affect the spread of *A. americanum* velocity ($IQR_{trans}$ = 0.356 cm/s). Weak, non-significant reductions were observed with both repellents for *I. scapularis*.

**3.2.2 Tick detachment.** Exposure to metofluthrin and transfluthrin did not generally affect the tendency of ticks to detach from their sticks (Fig 6). The sole significant difference was observed with metofluthrin against *I. scapularis*, where 33% (6/18) of ticks detached compared to 0% (0/18) of controls, P = 0.019. The size of this effect, though non-negligible, was not large enough to translate into a biologically notable measure when considering repellency standards (φ = 0.44).

**3.2.3 Climbing deterrence.** A significant climbing deterrence was observed in both treatment groups for all three species (Fig 7). 13% (2/15) of control *D. variabilis* were not deterred in comparison to 81% (17/21) of metofluthrin-exposed (P < 0.001, φ = 0.67) and 75% of transfluthrin-exposed (P < 0.001, φ = 0.60). A similar effect was observed in *A. americanum* trials. 13% (2/15) of controls were not deterred compared to 73% (11/15) of metofluthrin-exposed (P < 0.001, φ = 0.66), and 67% (12/18) of transfluthrin-exposed (P = 0.002, φ = 0.54). Metofluthrin performance in *I. scapularis* was comparable to that of *A. americanum* and showed 72% (13/18) deterrence when compared to 1/18 (6%) of controls (P < 0.001, φ = 0.68). Transfluthrin exposure in *I. scapularis* was associated with the weakest deterrence, though significant, with 50% (9/18) ticks having been deterred (P = 0.003, φ = 0.50).

**3.2.4 Pseudo-questing (P-Q) duration.** Control trials showed that the ticks of each species spent most of the trial at the top of the stick or climbed from the top of the stick to the top face of the chamber to remain (Fig 8). Metofluthrin and transfluthrin showed very similar reductions in *D. variabilis* pseudo-questing (P-Q) duration ($Mdn_{meto}$ = 0 sec.; $Mdn_{trans}$ = 0 sec.) when compared to controls ($Mdn_{control}$ = 495 sec.) (P < 0.001, r = 0.76). The spread of

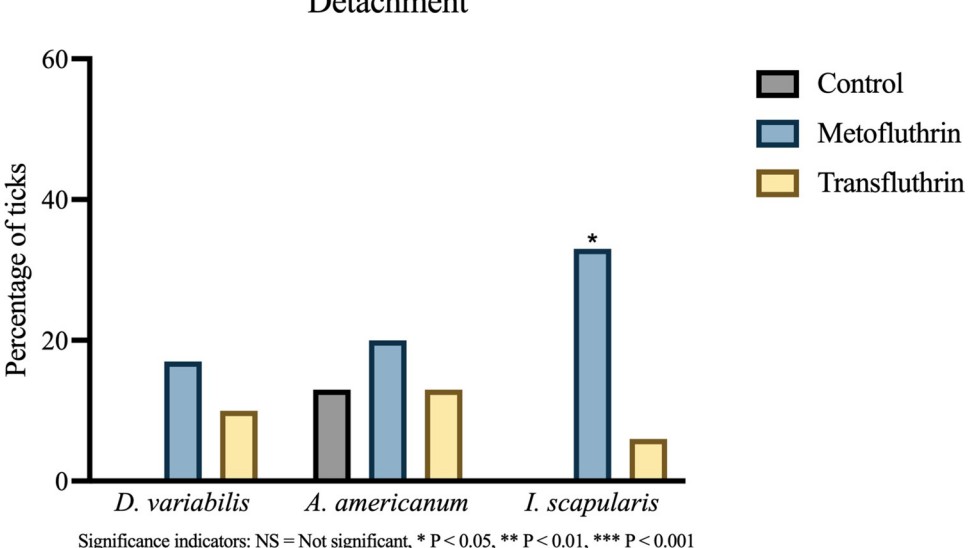

**Fig 6. Comparison of detachment.** The proportion of ticks that detached from their stick was assessed and shown as a percentage for each group. All ticks were included in the analysis. Only exposure to metofluthrin was associated with a small, significant increase in detachment in *I. scapularis*. Significance indicator: NS = Not significant, * P < 0.05, ** P < 0.01, *** P < 0.001.

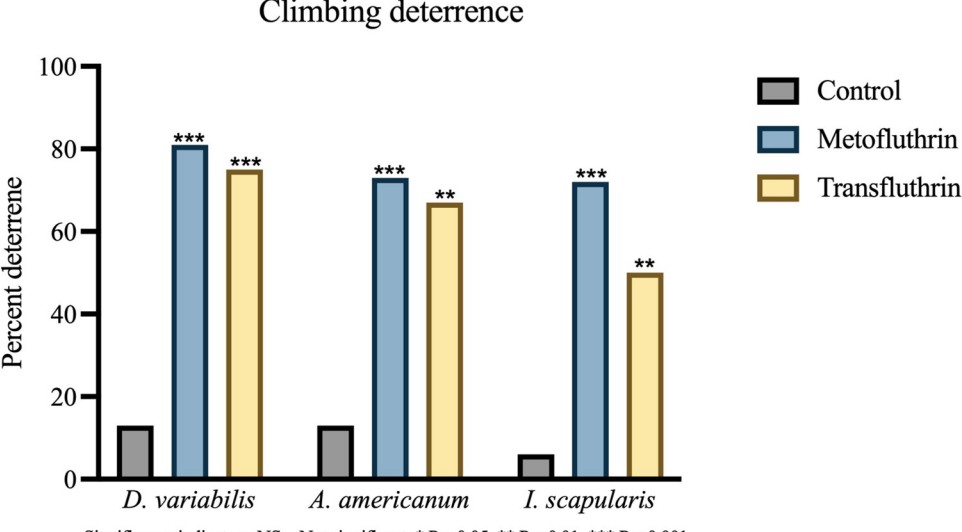

**Fig 7. Comparison of climbing deterrence.** The proportion of ticks deterred was described as a percentage. All ticks were included in the analysis. Significant deterrence was observed with transfluthrin and metofluthrin in all species. Significance indicator: NS = Not significant, * P < 0.05, ** P < 0.01, *** P < 0.001.

metofluthrin-exposed ticks was however more concentrated around the median (IQR = 71 sec.), in comparison to the similarly spread control and transfluthrin groups (IQR$_{control}$ = 171 sec., IQR$_{trans}$ = 189 sec.).

Metofluthrin-exposed *A. americanum* showed a reduction in P-Q duration comparable to *D. variabilis* ($Mdn_{meto}$ = 0 sec., $Mdn_{control}$ = 505 sec., P < 0.001, r = 0.76). A slightly weaker,

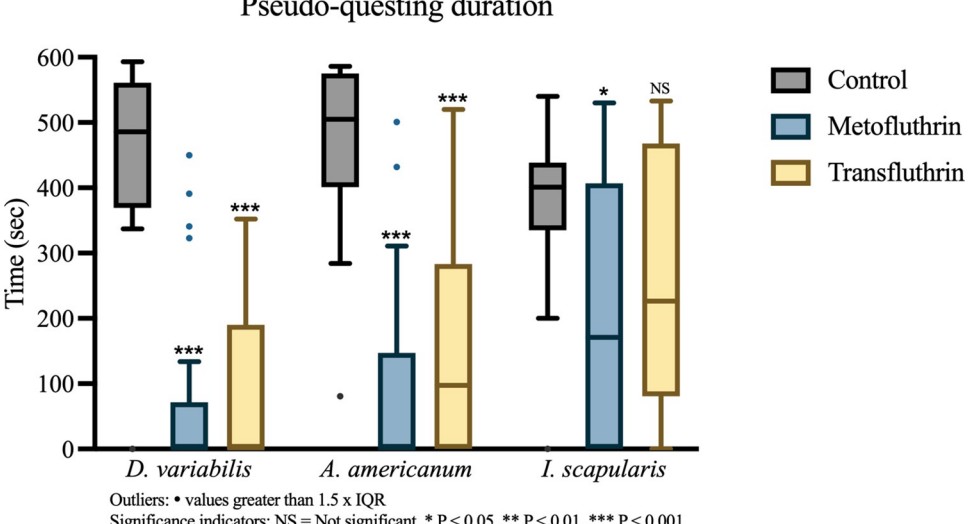

**Fig 8. Pseudo-questing duration.** Time that ticks spent above the 27 cm mark on the stick was considered time pseudo-questing (P-Q). Boxplots showed pairwise comparisons of repellent trials and controls, with cumulative P-Q duration (in sec.) of ticks in each group shown on the y axis. Significant reductions in P-Q duration were observed in both repellent groups in *D. variabilis* and *A. americanum*. *I. scapularis* showed a small, significant reduction in the presence of metofluthrin, but transfluthrin was not associated with a significant change. Outliers: • values greater than 1.5x IQR. Significance indicator: NS = Not significant, * P < 0.05, ** P < 0.01, *** P < 0.001.

but still significant reduction was seen in the transfluthrin group ($Mdn_{trans}$ = 98 sec., P < 0.001, r = 0.63). Contrasting the narrowing effect on dispersion seen against *D. variabilis*, the spread of P-Q duration in the *A. americanum* metofluthrin group (IQR = 147 sec.) was like that of the controls (IQR = 174 sec). Transfluthrin exposure in contrast increased P-Q duration spread (IQR = 283 sec.).

Reductions in P-Q duration in treatment groups were much smaller in *I. scapularis* when compared to *D. variabilis* and *A. americanum*. A weak, significant reduction was observed in metofluthrin trials ($Mdn_{control}$ = 401 sec., $Mdn_{meto}$ = 171 sec., P = 0.016, r = 0.40) while that observed with transfluthrin was not significant. Like the other species, *I. scapularis* controls were moderately spread (IQR = 104 sec.). Metofluthrin and transfluthrin exposure increased the spread of the groups ($IQR_{meto}$ = 407 sec., $IQR_{trans}$ = 387 sec.). Some ticks were relatively unaffected, having a P-Q duration comparable to that of controls, while others were observed to be more deterred.

## 4. Discussion

Significant effects were observed within each repellency parameter. Observations varied in strength by repellent group and by species. *D. variabilis* was generally more affected in each parameter and *I. scapularis* the least, except for in the detachment analysis, where the only significant difference with exposure occurred with metofluthrin in *I. scapularis*. Metofluthrin overall was associated with stronger performances within each indicator than transfluthrin. Both spatial repellents, however, showed the ability to deter climbing and pseudo-questing as well as distort tick movement as seen with reductions in velocity. It is important to consider different perspectives when measuring tick repellents, as they may work through different biomechanisms and therefore show their effects in different ways. Some measures will, however, be more descriptive than others. The most important parameter defined in the present study was climbing deterrence, which integrated several behaviors, however each parameter pertained to the host-seeking process in a different way.

The CFD simulation indicated the formation of a discernible vertical concentration gradient in the test chamber. This understanding allowed tick behavior to be connected to exposure. Deterrence of natural behavior to climb up was shown in several ways to be immediately affected upon tick placement into the chamber, where the concentration was the highest. Some remained immobile here for the entire trial. Further, significant reductions in velocity were seen as soon as ticks began climbing. The lack of tick movement opposing the concentration gradient indicated that the repellents did not act as a movement barrier but instead immediately disrupted favorable movement patterns aimed at the top of the box through an intoxication-like biomechanism. From this, assay changes would need to be made to assess the use of these repellents in a protective barrier role. Ticks would need to be placed in the chamber and allowed to climb into the repellent cloud. This can be done by reducing the induction time prior to tick introduction, increasing the height of the climbing sticks and chamber, and modifying the position of the CRD. Dose-stratified studies would also be required for further interpretation of the concentration gradient observed and to identify minimum and optimal concentrations for each spatial repellent, as the effective concentrations may differ from those used in the present study.

Host-seeking for ambushing ticks occurs through a process called questing, where they climb to the top of tall grass or other objects, hold their front legs out, and await a passing host on which to grab [53]. Once on host, they navigate upwards in search of a feeding site. Vertical repellency assays simulate this behavior, exploiting the tendency of ticks to climb [19, 38, 54–56]. In nature, a tick would be required to climb vertically without detaching from their

climbing surface–whether this be on grass or host. The proportion of ticks deterred from climbing could therefore apply to both settings. Detachment is another described indicator of inhibition in the host-seeking and feeding behavior of ticks [23]. If a tick detaches, it is not feeding or transmitting disease, whether this be from the top of grass it is questing on, while walking on the host, or early in the feeding process with its mouthparts inserted. In the context of spatial repellency evaluation, this could be evidence of the spatial repellent affecting the tick to the point where it could not function normally.

Questing behavior is naturally influenced by abiotic factors, including light, humidity, and temperature [57]. Ticks also have a limited supply of energy that once depleted cannot be restored. Therefore, ticks balance energy expenditure with the risks of questing by concentrating activity around conditions favorable to host-seeking and avoiding mortality [58]. Assuming ticks can differentiate conditions appropriate for questing from those that are not, it may be hypothesized that a reduction in questing time or increased immobility at lower heights may be evidence of questing deterrence.

Velocity, though unable to serve as a direct measure of repellency, can provide a measure of movement changes in the presence of spatial repellents. Observations in the present study showed this with reductions in velocity immediately when exposed. We can also observe differences innate to tick species, where *A. americanum* are naturally more fast-moving than *D. variabilis* and *I. scapularis*. Those that did not move at all in the trial, though not representing a statistically significant subset, may also represent ticks that were so affected by the high concentration of exposure that they were unable to proceed up the stick.

There are no spatial repellent products currently employed to protect from tick bites. Synthetic pyrethroids (metofluthrin, transfluthrin, allethrin) are used in passive CRDs (OFF! Clip-On Device, Personal Insect Repellent Kit [PIRK], ThermaCELL) as spatial repellents to protect against mosquitoes and other insects [40–42]. Recently, investigations into tick response to these devices have demonstrated varying spatial repellent effects and acaricidal activity in several species of ticks. *I. holocyclus* adult, female ticks have exhibited susceptibility to the OFF! Clip-On Device (31.2% metofluthrin) and ThermaCELL (21.9% allethrin), with exposures associated with spatial repellency and post-exposure mortality in the laboratory setting [40]. In a more natural environment, however, these effects were diminished. [40]. This shows the importance of studies performed in the natural setting, where external factors may impact spatial repellency. Other studies have shown similar effects with ThermaCELL, OFF! Clip-On Device, and PIRK (15.6% transfluthrin) against different life stages of I. *scapularis*, *A. americanum*, *and D. variabilis* in laboratory conditions without host-cues [41, 42]. Metofluthrin and transfluthrin in these experiments were associated with varying levels of mortality and spatial repellency between species. Neither repellent was associated with any mortality, during or after exposure. Further, the species-level susceptibility to the repellents appeared contrary to what has previously been demonstrated, with *D. variabilis* appearing the most affected and *I. scapularis* the least in our study. Consistent results however were shown with deterrence and observed behavior otherwise. Life stages, experimental conditions, tick sourcing, repellent formulation/release rate/induction time, and experiment duration among other factors must be taken into consideration when comparing the results of spatial repellent studies.

Chemoreception in ticks occurs through gustatory and olfactory mechanisms that are not well-understood [59]. A repellent biomechanism may work through one or both processes but tick repellent studies do not (or are not often able to) determine which is (are) at play [23]. The Haller's Organ is a complex sensory organ found on the tarsal segment of the front legs and is the basis of tick olfaction [53, 60–62]. It is comprised of two major structures: an anterior pit that detects humidity and a capsule fitted with host-cue sensing, physiologically diverse olfactosensilla. The response to repellents, including DEET, has been shown to diminish with the

occlusion or surgical removal of the Haller's Organ, suggesting that the Haller's Organ plays a role in the repellent response [60, 63]. Conversely, ticks have shown equal response to other repellents including phenethyl alcohol (PEA) with the Haller's Organ removed–alluding to other structures playing important roles in repellent chemoreception [63]. Considering PEA, removal of the pedipalps impairs repellent response. Further research is needed to characterize the molecular mode of action in tick olfaction and gustation to complement the analysis of novel tick-targeting repellents.

Although this represents a knowledge gap at the molecular level, it has been extensively demonstrated that tick behavior and response to biochemical stimuli can vary by species and life stage. Ticks of two species may show differences in responses or susceptibility to the same stimulus, under the same assessment conditions [64, 65]. This may be attributed to the size of the species or morphological differences in gustation and olfaction structures between species. Another important consideration pertains to the observable behavior of some species. Aggressive, fast-moving species, such as *A. americanum*, are often thought to produce a general underestimation of true susceptibility to repellents due to their higher speed and agility [66]. For instance, they may "blow past" a thin treatment line or barrier before they can even sense it.

Laboratory-reared ticks may also respond differently to repellents than those to which humans would be exposed. These differences may occur due to variation in age (ticks in the environment will have a more heterogeneous distribution of age) and genetic differences, highlighting a need to consider natural variation in future investigations [67]. The laboratory-reared ticks used in the present study were pathogen-free. It has been demonstrated that pathogens can modify the phenotype of their arthropod vector hosts in ways that benefit the pathogen. Some of these mechanisms increase vector competence by modifying vector behavior [68].

Considering insects, infections of *Plasmodium* spp. in *Anopheles* mosquitoes and *Leishmania* spp. in *Lutzomyia* sandflies are associated with molecular changes in olfactory mechanisms that modify host-seeking ability [69]. Similar mechanisms have not yet been described in ticks, but activity differences have been shown between pathogen-infected ticks and pathogen-free ticks that are relevant in the context of repellency [70–72]. *Borrelia burgdorferi* and *Anaplasma phagocytophilum*, the disease-causing pathogens for Lyme disease and anaplasmosis, have been shown to increase questing duration and walking activity in *I. scapularis* and *I. ricinus* adults and nymphs [70, 71]. Further, tick-borne encephalitis virus has been shown to increase DEET tolerance of *I. ricinus* [72]. Repellency testing should proceed to account for these factors to provide appropriate estimates for the level of protection humans may receive with exposure to pathogenic ticks.

The external validity of *in vitro* evaluations of repellency is limited in the absence of host cues and natural conditions. Preliminary assessments of repellent function are best performed in the laboratory setting for safety and economic purposes; however, external factors may impact protective effects as it is currently unknown how wind or human movement around the CRD may affect the protective barrier. Other factors, such as temperature and humidity, may affect the extent to which the target vector responds and disrupt the gradient barrier with changes in vapor pressure. Because the stability of the volatilized concentration serves as the basis of a protective barrier with spatial repellents, tests in natural conditions are an essential step in product development. The present study represents the earliest stages in repellent testing and a point from which to scale. Future work should assess effects in larger, more representative populations and in more natural environments. In practical use, it would be hypothesized that these factors would not affect the spatial concentration around the device enough to reduce the repellent effect, allowing the CRD to serve as a wearable barrier to tick bites. This, however, has yet to be effectively demonstrated.

Management practices and personal protection lie at the forefront of tick bite prevention, as a "magic bullet" solution for tick-borne disease does not yet exist [8, 53, 73]. At the individual level, repellent use is one of the most important tactics for protection. DEET and permethrin are the most widely used repellents in tick protection. Due to limitations with safety concerns, imperfect human agreeability, and inconsistency of protective duration and efficacy across tick species, new repellents are still needed [23]. The ideal repellent requires an active ingredient formulated to provide long-lasting, efficacious protection against a range of disease-transmitting vectors in a safe, pleasant formula for human use [53, 74]. Spatial repellency research to date has set a path for future work which may provide humans with novel, efficacious spatial repellents to target ticks. The present study has identified several parameters which may be used to assess spatial repellency, considering the natural questing behavior of ambushing ticks, and demonstrated significant effects on tick behavior with transfluthrin and metofluthrin based on these parameters.

## Supporting information

**S1 Fig. CFD simulation mesh: Front, right, bottom, isometric views.**
(PDF)

**S2 Fig. Detailed view of the mesh used around the device, isopropanol, and repellent source in CFD simulation.**
(PDF)

**S1 Table. Properties and input flow rates for transfluthrin, metofluthrin, and isopropanol.**
(PDF)

**S1 Appendix. CFD and GCMS.**
(DOCX)

**S1 Data. Raw data from climbing trials.**
(XLSM)

## Acknowledgments

The authors would like to thank Mr. Kevin Smith as representative of Bayer Environmental Sciences for providing the technical transfluthrin to conduct the research. His enthusiasm and corporate contribution are greatly appreciated.

This article reports the results of research only. Mention of a proprietary product does not constitute an endorsement or a recommendation by the authors, USDA, or DoD for its use. The USDA is an equal opportunity provider and employer.

## Author Contributions

**Conceptualization:** Sebastián D'hers, Noel M. Elman, Stephen M. Rich.

**Formal analysis:** Eric L. Siegel, Marcos Olivera, Esteban Martinez Roig, Sebastián D'hers.

**Funding acquisition:** Eric L. Siegel, Noel M. Elman, Stephen M. Rich.

**Investigation:** Eric L. Siegel, Melynda Perry, Sebastián D'hers, Noel M. Elman, Stephen M. Rich.

**Methodology:** Andrew Y. Li, Sebastián D'hers, Noel M. Elman, Stephen M. Rich.

**Project administration:** Stephen M. Rich.

**Resources:** Stephen M. Rich.

**Supervision:** Noel M. Elman, Stephen M. Rich.

**Validation:** Eric L. Siegel.

**Visualization:** Marcos Olivera, Esteban Martinez Roig, Sebastián D'hers.

**Writing – original draft:** Eric L. Siegel, Sebastián D'hers, Noel M. Elman, Stephen M. Rich.

**Writing – review & editing:** Eric L. Siegel, Melynda Perry, Andrew Y. Li, Sebastián D'hers, Noel M. Elman, Stephen M. Rich.

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
