## [Decision Letter · Decision Letter 0]

26 Jul 2022

PONE-D-22-14066A new paradigm for personal protection against ticks: efficacy of spatial    repellents to reduce host seeking activities in three major tick species of medical importancePLOS ONE

Dear Dr. Rich,

Thank you for submitting your manuscript to PLOS ONE. After careful consideration, we feel that it has merit but does not fully meet PLOS ONE’s publication criteria as it currently stands. Therefore, we invite you to submit a revised version of the manuscript that addresses the points raised during the review process.

 The reviewers were generally positive about the potential for this work but had concerns about the clarity of the manuscript. Please address the comments carefully and provide the level of detail the reviewers have asked for.

We look forward to receiving your revised manuscript.

Kind regards,

Catherine A. Brissette, Ph.D.

Academic Editor

PLOS ONE

Journal Requirements:

“This study was supported by the DoD Tick-borne Disease Program from the Congressional Directed Medical Research Program, grant # W81XWH1920028.”

Reviewers' comments:

Reviewer's Responses to Questions

**Comments to the Author**

1. Is the manuscript technically sound, and do the data support the conclusions?

Reviewer #1: Partly

Reviewer #2: Partly

Reviewer #3: Yes

2. Has the statistical analysis been performed appropriately and rigorously? 

Reviewer #1: Yes

Reviewer #2: I Don't Know

Reviewer #3: Yes

3. Have the authors made all data underlying the findings in their manuscript fully available?

Reviewer #1: Yes

Reviewer #2: Yes

Reviewer #3: Yes

4. Is the manuscript presented in an intelligible fashion and written in standard English?

Reviewer #1: Yes

Reviewer #2: No

Reviewer #3: No

5. Review Comments to the Author

Reviewer #1: This study presents a unique bioassay that evaluates the response of three medically important tick species in the US to various volatilized AIs. As noted in the manuscript, development of such bioassays is an important first step in evaluating candidate AIs for use as spatial repellents. However, it is difficult to compare actual efficacy among the compounds used because only a single concentration was applied per compound (though clearly the assay provides a concentration gradient). I would suggest rethinking the title of the manuscript. The findings are essentially a description of a novel bioassay, which is very significant, but they do not actually show the “efficacy of spatial repellents to reduce host-seeking activity” as a new paradigm for personal protection. The assay seems effective at screening different AIs, but wonder if a concentration gradient should have been used for each AI. It is possible that the optimal concentration is not the same across compounds and this should be noted in the Discussion as a limitation. I’m not sure how transferable this information is to a natural setting. I realize scaling efficacy trials is a goal for future studies, but the title of the paper suggests a more definitive answer would be presented herein. This would also be worth a bit more discussion.

Overall, this was a clearly written manuscript describing a novel bioassay. A few minor points are listed below.

Line 205: Why were ticks that did not move during the trials omitted? Maybe I’m not understanding the methods as written, but it sounds like you first screened ticks in the absence of AIs to ensure they were willing to ascend sticks. If that’s the case, why omit the ones that didn’t move in the presence of an AI? Seems lack of movement would be a valid response if the tick is at the bottom of the stick and chooses not to ascend away from the repellent? Please provide additional explanation for why these were excluded and what immobility might imply about the efficacy of the compound.

Results (and Tables): If the data are analyzed using non-parametric statistics due to a lack of normality, why present the results as means and standard deviations, rather than medians and ranges?

Reviewer #2: This manuscript presents results of experiments that tested the impacts of exposure to 4 repellents (nootkatone, DEET, transfluthrin, metafluthrin) on the behavior of 3 species of ticks, Amblyomma americanum, Dermacentor variabilis, and Ixodes scapularis. Each species is evaluated for the impacts of spatial exposures rather than contact responses. The authors also describe the bioassay apparatus and developed several metrics to assess behavior, using video capture as a tool for these analyses.

Major weaknesses include:

1. The idea of non-contact spatial repellency is not a new paradigm. Others have considered and reported evaluation of this for ticks. Ticks Tick Borne Dis. 2017 Oct;8(6):837-849.doi: 10.1016/j.ttbdis.2017.06.010. Epub 2017 Jul 1.

2. Simulations of spatial gradients in the chamber were only done with metafluthrin and transfluthrin. Why weren’t DEET and Nootkatone assessed?

3. The results are confounded because some of the ticks were collected in the environment (wild-caught) and some were purchased from a rearing facility in Oklahoma. There is no description of the relative proportions of the wild-caught vs colony ticks used for the assays and no validation that colony ticks performed in the same way as wild ticks. This matters because the genetic background of the OSU ticks is unknown. We do know that southern versus northern Ixodes scapularis ticks do not behave in the same way and this has a genetic component (Arsnoe et al. 2015). Field caught ticks may also have other conditions that could affect behavior, including infection status and physiological age. A better study design would be to compare an adequate sample of ticks from the lab colony with a similar number of ticks collected from a field site. Mixed samples are not acceptable unless adequate testing has been done to assure the colony and wild specimens behave identically.

4. No variance estimates for some of the metrics are provided. Did ticks perform the same between the 4-6 replicates?

5. Some of the reported outcomes lump all species together (Table 1 and in-text). This is not appropriate given the differences that were observed.

6. The authors do not clearly describe how they expect these materials to be used. Spatial repellents for mosquitoes work for malaria control because mosquitoes enter homes and bite humans there so repellents can protect entry points of the homes. Spatial repellents for humans doing yardwork or hiking don’t work to reduce mosquito bites because the volatile cloud produced by a personal device isn’t stable. A clearer description of how they envision spatial repellents working to prevent tick bites would be useful for readers.

The manuscript is not well-written or well-edited. There are multiple places where the wording is incorrect, unclear or so imprecise as to be uninterpretable. Errors of grammar and formatting are common throughout this manuscript. Some of the many examples of these problems include:

Abstract

1. Line 30: “of Integrative Vector Management (IVM) program”. Either “an IVM program” or “IVM programs”

2. Line 30: The accepted term is Integrated Vector Management, not integrative.

3. “…despite the speculated value of volatilized chemicals in control systems.” Who is speculating? A reference is needed. Or consider “potential value”.

4. “…novel vertical climb assay”. What makes the assay novel? Others have used vertical sticks in arenas to study tick questing behavior. Others have filmed vertical movement behavior and analyzed it. Consider simply describing the assay without the qualifier.

5. Line 41: “…and changing in tick climbing behavior when compared controls“ Grammatically incorrect.

6. Line 45: “slightly weaker effects “. This does not accurately reflect observations. DEET and Nootkatone appeared to be more effective for I. scapularis. Compare with lines 372-376 “Exposure to all four AIs was associated with significant reductions in pseudo-questing tendency in D. variabilis and A. americanum. This association was strongest with metofluthrin and transfluthrin in both species. In I. scapularis, nootkatone showed the strongest effect, however DEET and metofluthrin showed smaller, significant reductions. Only transfluthrin was not associated with a significant reduction.”

Introduction

1. Line 55-57. “The prevalence of these zoonotic diseases has increased recently due to shifts in host population dynamics, particularly with the white-tailed deer, that affect tick population size [2]. “ Many would dispute this assertion that white-tailed deer are the primary drivers of recent increases in tickborne disease prevalence. Many studies have failed to find a strong correlation between deer abundance and tickborne diseases like Lyme disease. Human behavior and exposure is a key issue. Defining "recent" could help place this in context-last hundred years?

2. Line 57-58: “Targeting of live arthropod populations…” The word “live” is unnecessary. The rest of the sentence could be restated as “an important component of integrated vector management program is vector management”. I’m struggling to understand what component of IVM would not involve targeting the vectors, whether they are on or off hosts.

3. Line 60: “source reduction” has a specific meaning in vector biology-the elimination of breeding sites for mosquitoes (or vectors without aquatic life phases). Using the term to mean all types of environmental control as opposed to personal prevention muddies the definition and should be avoided.

4. Line 62-63: “…while personal protective methods seek to reduce risk to individual humans through smaller-scale, personal chemical application.” Again, this statement lacks precision. Most tick biologists consider personal protective methods to be broader than just personal chemical application. Such methods also include tick checks, showering soon after potential exposure, tucking pants into socks, and putting clothes in a hot dryer after being outside.

5. Line 73-75. “The host-seeking behavior and ecology of ticks, however, challenges the applicability of these repellent biomechanisms that are traditionally used to combat more agile, flying arthropods.” The authors should be specific about why ticks would not be expected to exhibit inhibition of attraction to a host, irritancy, or intoxication as a result of exposure to a repellent, simply because they don’t fly. The DEET example shows that the mechanisms may change, not that they are not applicable.

6. Line 80. “Other known pyrethroids….” This paragraph/sentence structure makes it sound like the authors are saying DEET is a pyrethroid.

7. Line 82. Again, the use of “source reduction” does not align with practices in the field. ULV is mostly used for adulticiding, not for source reduction (targeting larval sites). The next sentence makes clear that it is targeting adults with ULV that the authors are referring to.

8. Line 86-88. Please clarify the statement to reference lack of standardized assays for spatial repellents as opposed to contact repellents and cite the papers that do define standard assays for assessing contact repellents for ticks. The US EPA and the UK both have defined acceptable assays for regulatory processes. You should also cite the papers that have looked at spatial repellency for ticks, e.g. for permethrin treated clothing.

Results

1. “Few A. americanum detached from their sticks in control trials (2) however only once reaching the top, and no I. scapularis or D. variabilis in theirs. (Table I).” This sentence needs rewriting for clarity.

Discussion.

1. Line 367-368. “Thus, they must use this supply wisely” Line 396. “…..desired location”. Avoid words like ‘wisely’ and ‘desired’ in describing tick behavior.

2. Line 383-385. “There were several occurrences of large changes in the distance ticks traveled. The greatest of which were with metofluthrin and transfluthrin, which reduced the displacement of all three species.” Sentence fragment.

3. Line 392-394. “The reduction in velocity shown by metofluthrin and transfluthrin in D. variabilis and A. americanum could be evidence of visual effects of AI interference in ticks’ natural ability to move.” Could the authors clarify how AI would interfere with tick vision?

4. Line 399-412. How would a tick experience the gradient? Considering size of a tick relative to volume of the container and behavior of the AI materials, would you expect a tick at the bottom to sense a gradient?

To reiterate, these are just examples of problematic statements, language, and editing. There are many more. Much of the paper needs a major overhaul.

Reviewer #3: PLOS ONE

Manuscript No. PONE-D-22-14066

Authors: Siegel EL, et al.

Many of the explanations and descriptions given in this manuscript were weak and appeared rather hurriedly done. The title of the manuscript claims that this work represents a new paradigm in evaluating spatial repellents to ward off tick parasitism. Such a paradigm will be welcomed by many. But if this work is to truly present a new way forward, then everything – from describing and illustrating the apparatus to providing methods of calculating and analyzing outcome variables – needs to be explained in a more clear and concise manner. Some of the ways this can be done are listed below.

There were 6 behavioral outcomes (i.e., dependent variables) used to assess the spatial repellency of 4 different volatiles. Some of the variables (e.g., climbing success versus climbing height reduction) appeared to be highly interdependent. The manuscript would increase its usefulness to other vector biologists if authors could either rank the variables or make a recommendation, based on their experience, on which of the six might are the most useful and meaningful to employ when testing volatiles for spatial repellency of host-seeking ticks.

INTRODUCTION.

Lines 102-103. This is the last sentence in the final paragraph of the Introduction where authors typically state the research objective of their study. But the sentence construction here is jagged. There are several dependent clauses all strung together at the end of an already long sentence. This tends to muddle the meaning. Suggest condensing or break it into two sentences.

MATERIALS & METHODS

Description of the apparatus needs improvement. For example:

- Line 126. Is a ‘chemical-emanating device’ the same as a ‘controlled release device’? If so, then please refer to the device using a single, consistent term. Otherwise, it is confusing.

- Line 129. Before describing the behavioral test chamber, it is appropriate to provide a description of the controlled release device.

- Figure 1. Please provide dimensions of the controlled release device.

- Line 135. “placement of the active ingredient” – shouldn’t that be “placement of the CRD containing the AI being tested”?

- Line 138. Please state the composition of the sticks. Wooden sticks? Plastic sticks? Paper sticks?

- Figure 2 legend; Line 148. Change the word ‘device’ to ‘controlled release device [CRD]” so that it matches what is shown in the diagram.

- Line 159. Again, is the term ‘emanating device’ the same thing as the ‘controlled release device’? Please use consistent terminology throughout.

- Line 182 – Video Tracking. EthoVision is a software program. Please provide brief description of the hardware and conditions used – e.g., camera, level of magnification if any, etc. This can be provided as a Supplement section.

RESULTS – 6 outcome variables. WHICH IS THE MOST IMPORTANT ??

Tables – keep the order of tick species consistent – e.g., Amblyomma – Dermacentor – Ixodes. Mixing the order all around from one table to the next adds unnecessary confusion for the reader.

Figures. – similarly, keep the order of the test compounds consistent from table-to-table and figure-to-figure (e.g., Fig. 5 differs from Figs. 6-8).

OUTCOME VARIABLES

Tick Detachment – Table 1 Authors should point out that none of the 4 AI’s significantly altered detachment for A. amblyomma. The significant effect of metofluthrin on tick detachment was due solely to the response of D. variabilis and I. scapularis ticks. Thus, the statement on line 416 is incorrect. With A. americanum and metofluthrin, detachment in the control group was 2/15. Detachment in the metofluthrin group was 3/15 (Table 1). No difference.

Mean displacement – Table 2 - Data were not normally distributed (lines 201-202). Therefore, values should be presented as geometric means and 95% CL, not arithmetic means + SD.

Climbing height reduction – Fig. 5 – What do the numbers signify atop each histogram ? Percentages? They should also indicate the sample size. Line 192. Climbing height reduction needs to be more fully explained – e.g., how were heights of AI-exposed ticks ‘normalized’ to non-exposed control ticks? Was there some kind of mathematical formula used? If so, please provide.

Climbing Success – Fig. 6 – what is the difference between Climbing Success and Climbing Height Reduction. This should be explained clearly because the pattern of histograms (Fig. 5 & Fig. 6) appear nearly identical for two of the three tick species. Are you showing the same thing or something different in Figs. 5 & 6. This should be made clear.

Much of the text in Figure legends – Fig. 6, 7, 8 – should be taken out of the figure legends and placed in the appropriate place within the text of Results section.

FIGURES 3, 4, 7 & 8. The axes labels are too small and need to be enlarged to be legible.

FIGURE 3a & 3b. No exaggeration, I had to use a magnifying glass to read the labels on the y-axes. The x-axis was not labelled at all. Where is the CRD placement in relation to the configuration of this chart? Presumably at top left corner for larger box, but at top right for smaller box? This needs to be clarified.

FIGURE 7. Likewise, the font chosen to express significance level is too small.

TABLE 3. Use a footnote or some other way to denote what the asterisks signify. Also, use a zero to the left of the decimal point, otherwise a reader might overlook the small little dots (decimal points) in a table containing 12 rows of data.

Line 204. States that all ticks were included. But lines 174-175 states that some ticks were excluded. Which is correct?

DISCUSSION

Line 347 – What is meant by the phrase “build on the shortcomings”? Building anything new based on the defective ideas/practices of the past does not a seem like the best approach for progress. Is this really what the authors meant to say?

Line 408 – “pushing ticks to continue questing for a safer place”? Not sure what this means. Is there any evidence to support the notion that ticks are sentient enough to sense danger? Perhaps it is better to frame this in terms of a mechanistic response to volatiles

Lines 462-464. The ideas contained in this sentence are not relevant to a discussion of spatial repellents distributed by controlled release devices. To avoid confusing the use of personal repellents with spatial repellents, this sentence should be removed.

MINOR GRAMMATICAL & MISCELLANEOUS

Line 65. Add word “repellent” between the words ‘spatial’ & ‘compound’.

Line 67. Change ‘are’ to ‘is’; or delete phrase “A new generation of …”.

Lines 75 & 83. Change the word ‘arthropods’ to ‘insects’. Insects are the only arthropod capable of flight. Therefore flying arthropods are by default, flying insects.

Line 109. :.. make test 30% test …” Remove the 1st test?

Line 115. State the species of wild ticks that were collected in North Amherst, MA used to supplement the OSU colony ticks.

Line 192. Something is missing in this sentence.

Line 239. Species name should be written as lower case. Watch out for auto-correct.

Lines 237 – 247. Authors switch from past tense to present tense, then back to past tense. Suggest sticking with just past tense in reporting results.

There are 13 abbreviations used throughout this manuscript. Are they all necessary? If you only use a term once or twice (e.g., ULV) why is it necessary to mark it with an abbreviation? Likewise, it is never good to begin a sentence with an abbreviation. Just as the first word in a sentence should be capitalized, the subject of a sentence should be spelled out.

6. PLOS authors have the option to publish the peer review history of their article (what does this mean?). If published, this will include your full peer review and any attached files.

Reviewer #1: No

Reviewer #2: No

Reviewer #3: No

---

## [Author Response · Author response to Decision Letter 0]

9 Sep 2022

Please find the authors' response to individual reviewer comments in the attached file "Response to reviewers." Responses to individual comments are also pasted below: 

Reviewer #1: This study presents a unique bioassay that evaluates the response of three medically important tick species in the US to various volatilized AIs. As noted in the manuscript, development of such bioassays is an important first step in evaluating candidate AIs for use as spatial repellents. However, it is difficult to compare actual efficacy among the compounds used because only a single concentration was applied per compound (though clearly the assay provides a concentration gradient). I would suggest rethinking the title of the manuscript. The findings are essentially a description of a novel bioassay, which is very significant, but they do not actually show the “efficacy of spatial repellents to reduce host-seeking activity” as a new paradigm for personal protection. The assay seems effective at screening different AIs, but wonder if a concentration gradient should have been used for each AI. It is possible that the optimal concentration is not the same across compounds and this should be noted in the Discussion as a limitation. I’m not sure how transferable this information is to a natural setting. I realize scaling efficacy trials is a goal for future studies, but the title of the paper suggests a more definitive answer would be presented herein. This would also be worth a bit more discussion. 

Overall, this was a clearly written manuscript describing a novel bioassay. A few minor points are listed below. 

1. Rethink the title of the manuscript. The findings are essentially a description of a novel bioassay, which is very significant, but they do not actually show the “efficacy of spatial repellents to reduce host-seeking activity” as a new paradigm for personal protection. 

The authors appreciate the feedback, agree with the reviewer, and feel the new title better represents the study. The title has been revised: “Spatial repellents transfluthrin and metofluthrin affect the behavior of Dermacentor variabilis, Amblyomma americanum, and Ixodes scapularis in an in vitro vertical climb assay.” 

2. If a concentration gradient should have been used for each AI. It is possible that the optimal concentration is not the same across compounds and this should be noted in the Discussion as a limitation 

The authors thank the reviewer for raising this valid point. This study is being treated as a preliminary assessment of both the repellents and the methodology. The concentrations used in the study are possibly not the lowest effective or optimal concentrations for use. The concentrations used in this study are similar to what has previously demonstrated repellency for the spatial repellents. Scaling concentration is a future goal once determining an effect was present. This has been noted as a limitation in the discussion. Please refer to the discussion section, paragraph 2, lines 397-403 for this updated information. 

3. I’m not sure how transferable this information is to a natural setting. I realize scaling efficacy trials is a goal for future studies, but the title of the paper suggests a more definitive answer would be presented herein. This would also be worth a bit more discussion. 

The authors thank the reviewers for noting this. It is hypothesized that the spatial concentration maintained by the controlled release device will be great enough to affect ticks in a more natural environment and serve as a barrier repellent. The discussion notes factors that affect spatial repellent function outside and these will need to be taken into consideration in future tests. It is unknown whether this will transfer to a natural environment, and this is the next step for testing. The title has been revised as to not overstate results. Further, please refer to the discussion section, lines 495-507 where this is further addressed. 

4. Line 205: Why were ticks that did not move during the trials omitted? Maybe I’m not understanding the methods as written, but it sounds like you first screened ticks in the absence of AIs to ensure they were willing to ascend sticks. If that’s the case, why omit the ones that didn’t move in the presence of an AI? Seems lack of movement would be a valid response if the tick is at the bottom of the stick and chooses not to ascend away from the repellent? Please provide additional explanation for why these were excluded and what immobility might imply about the efficacy of the compound. 

The authors thank the reviewer for raising points that were unclear in the previous version. The manuscript has been updated to better address these comments. Please refer to the materials and methods, section 2.5.1 (lines 189-199), for an updated and clarified description on the activity assessment. Further, the results, section 2.5.1 (lines 190-199), and discussion, lines 404-429) have been revised to address these concerns. Immobility should not have been understated. The methods have been revised to clarify that ticks that remained immobile have been considered in an updated analysis, where they are only excluded to evaluate tick velocity while moving along the stick. 

5. Results (and Tables): If the data are analyzed using non-parametric statistics due to a lack of normality, why present the results as means and standard deviations, rather than medians and ranges? 

The authors thank the reviewer for bringing up this issue with analysis and presentation. The velocity and pseudo-questing analyses have been updated where necessary with medians as a measure of central tendency and interquartile range as a measure of dispersion. Please refer to figures 5 and 8 for these updates as well as their corresponding results sections, 3.2.1 (lines 279-317) and 3.2.4 (lines 346-373). 

Reviewer #2: This manuscript presents results of experiments that tested the impacts of exposure to 4 repellents (nootkatone, DEET, transfluthrin, metafluthrin) on the behavior of 3 species of ticks, Amblyomma americanum, Dermacentor variabilis, and Ixodes scapularis. Each species is evaluated for the impacts of spatial exposures rather than contact responses. The authors also describe the bioassay apparatus and developed several metrics to assess behavior, using video capture as a tool for these analyses. 

Major weaknesses include: 

1. The idea of non-contact spatial repellency is not a new paradigm. Others have considered and reported evaluation of this for ticks. 

The authors thank the reviewer for the feedback. Prior research in spatial repellency in ticks has been better represented in the manuscript. Please refer to the introduction, lines 100-105, and the discussion section, lines 430-450 for further background on spatial repellency investigations in ticks with permethrin, metofluthrin, transfluthrin, allethrin, and DEET. 

2. Simulations of spatial gradients in the chamber were only done with metafluthrin and transfluthrin. Why weren’t DEET and Nootkatone assessed? The results are confounded because some of the ticks were collected in the environment (wild-caught) and some were purchased from a rearing facility in Oklahoma. There is no description of the relative proportions of the wild-caught vs colony ticks used for the assays and no validation that colony ticks performed in the same way as wild ticks. This matters because the genetic background of the OSU ticks is unknown. We do know that southern versus northern Ixodes scapularis ticks do not behave in the same way and this has a genetic component (Arsnoe et al. 2015). Field caught ticks may also have other conditions that could affect behavior, including infection status and physiological age. A better study design would be to compare an adequate sample of ticks from the lab colony with a similar number of ticks collected from a field site. Mixed samples are not acceptable unless adequate testing has been done to assure the colony and wild specimens behave identically. 

The reviewer raises a valid point regarding the simulations and the inclusion of field collected ticks. Nootkatone and DEET have been removed from the study. Further, only OSU ticks are included. Please refer to the discussion, lines 474-494 for further information comparing laboratory-reared ticks and flagged ticks in the context of tick repellency. For updated simulation discussion, please refer to lines 389-404. 

4. Some of the reported outcomes lump all species together (Table 1 and in-text). This is not appropriate given the differences that were observed. 

The authors thank the reviewer for raising this point. All results have been revised. Please refer to the results, section 3 for this information. Species are considered separetely. 

5. The authors do not clearly describe how they expect these materials to be used. Spatial repellents for mosquitoes work for malaria control because mosquitoes enter homes and bite humans there so repellents can protect entry points of the homes. Spatial repellents for humans doing yardwork or hiking don’t work to reduce mosquito bites because the volatile cloud produced by a personal device isn’t stable. A clearer description of how they envision spatial repellents working to prevent tick bites would be useful for readers. 

The authors thank the reviewer for this comment. Please refer to the discussion, line 504-507 for this information. It is hypothesized that the device will function as a wearable barrier to tick bites. This needs to be assessed in natural conditions before this claim can be made. 

6. The manuscript is not well-written or well-edited. There are multiple places where the wording is incorrect, unclear or so imprecise as to be uninterpretable. Errors of grammar and formatting are common throughout this manuscript. Some of the many examples of these problems include: Abstract 1. Line 30: “of Integrative Vector Management (IVM) program”. Either “an IVM program” or “IVM programs” 2. Line 30: The accepted term is Integrated Vector Management, not integrative. 3. “…despite the speculated value of volatilized chemicals in control systems.” Who is speculating? A reference is needed. Or consider “potential value”. 4. “…novel vertical climb assay”. What makes the assay novel? Others have used vertical sticks in arenas to study tick questing behavior. Others have filmed vertical movement behavior and analyzed it. Consider simply describing the assay without the qualifier. 5. Line 41: “…and changing in tick climbing behavior when compared controls“ Grammatically incorrect.6. Line 45: “slightly weaker effects “. This does not accurately reflect observations. DEET and Nootkatone appeared to be more effective for I. scapularis. Compare with lines 372-376 “Exposure to all four AIs was associated with significant reductions in pseudo-questing tendency in D. variabilis and A. americanum. This association was strongest with metofluthrin and transfluthrin in both species. In I. scapularis, nootkatone showed the strongest effect, however DEET and metofluthrin showed smaller, significant reductions. Only transfluthrin was not associated with a significant reduction.” 

The authors thank the reviewers for raising these issues with the abstract and introduction. The abstract has been rewritten entirely. The information has been presented to better reflect the study’s goals, results, and conclusions. 

Introduction 

7. Line 55-57. “The prevalence of these zoonotic diseases has increased recently due to shifts in host population dynamics, particularly with the white-tailed deer, that affect tick population size [2]. “ Many would dispute this assertion that white-tailed deer are the primary drivers of recent increases in tickborne disease prevalence. Many studies have failed to find a strong correlation between deer abundance and tickborne diseases like Lyme disease. Human behavior and exposure is a key issue. Defining "recent" could help place this in context-last hundred years? 

The authors appreciate the comment regarding the drivers of tickborne disease incidence. The introduction has been revised to better describe the human risk factors and ecological contributions to tick borne disease incidence. Please refer to lines 58-61 of the introduction. 

8 Line 57-58: “Targeting of live arthropod populations…” The word “live” is unnecessary. The rest of the sentence could be restated as “an important component of integrated vector management program is vector management”. I’m struggling to understand what component of IVM would not involve targeting the vectors, whether they are on or off hosts. 

3. Line 60: “source reduction” has a specific meaning in vector biology-the elimination of breeding sites for mosquitoes (or vectors without aquatic life phases). Using the term to mean all types of environmental control as opposed to personal prevention muddies the definition and should be avoided. Line 62-63: “…while personal protective methods seek to reduce risk to individual humans through smaller-scale, personal chemical application.” Again, this statement lacks precision. Most tick biologists consider personal protective methods to be broader than just personal chemical application. Such methods also include tick checks, showering soon after potential exposure, tucking pants into socks, and putting clothes in a hot dryer after being outside. Line 82. Again, the use of “source reduction” does not align with practices in the field. ULV is mostly used for adulticiding, not for source reduction (targeting larval sites). The next sentence makes clear that it is targeting adults with ULV that the authors are referring to. 

The authors thank the reviewer for raising these issues with terminology use and phrasing. The introduction has been re-written to focus on personal protection as opposed to environmental protection and therefore the source reduction information has been removed. Further information with personal protection has been expended in the introduction, lines 63-78. 

9. Line 73-75. “The host-seeking behavior and ecology of ticks, however, challenges the applicability of these repellent biomechanisms that are traditionally used to combat more agile, flying arthropods.” The authors should be specific about why ticks would not be expected to exhibit inhibition of attraction to a host, irritancy, or intoxication as a result of exposure to a repellent, simply because they don’t fly. The DEET example shows that the mechanisms may change, not that they are not applicable. 

The authors thank the reviewers for raising these points of confusion. This information has been revised, lines 86-90 for a better definition of repellency in ticks. 

10. Line 80. “Other known pyrethroids….” This paragraph/sentence structure makes it sound like the authors are saying DEET is a pyrethroid. 

The authors acknowledge the lack of clarity in this sentence. Please refer to lines 70-78 for updated information. 

11. Line 86-88. Please clarify the statement to reference lack of standardized assays for spatial repellents as opposed to contact repellents and cite the papers that do define standard assays for assessing contact repellents for ticks. The US EPA and the UK both have defined acceptable assays for regulatory processes. You should also cite the papers that have looked at spatial repellency for ticks, e.g. for permethrin treated clothing. 

The authors thank the reviewer for the suggestion. Please refer to lines 97-105 for expansion on these details. 

Results 

6. “Few A. americanum detached from their sticks in control trials (2) however only once reaching the top, and no I. scapularis or D. variabilis in theirs. (Table I).” This sentence needs rewriting for clarity. 2. Line 383-385. “There were several occurrences of large changes in the distance ticks traveled. The greatest of which were with metofluthrin and transfluthrin, which reduced the displacement of all three species.” Sentence fragment. 

The authors thank the reviewer for raising these points. The results section has been revised to account for these issues. 

12. Line 367-368. “Thus, they must use this supply wisely” Line 396. “…..desired location”. Avoid words like ‘wisely’ and ‘desired’ in describing tick behavior. 

The authors thank the reviewer for the comment. This information has been revised, please refer to lines 419-420 of the discussion. 

13. Line 392-394. “The reduction in velocity shown by metofluthrin and transfluthrin in D. variabilis and A. americanum could be evidence of visual effects of AI interference in ticks’ natural ability to move.” Could the authors clarify how AI would interfere with tick vision? 

The authors thank the reviewers for raising this issue. The information presented was not clear. This has been revised to reflect a more appropriate “observable” than “visual.” Please refer to the discussion, lines 424-430. 

14. Line 399-412. How would a tick experience the gradient? Considering size of a tick relative to volume of the container and behavior of the AI materials, would you expect a tick at the bottom to sense a gradient? To reiterate, these are just examples of problematic statements, language, and editing. There are many more. Much of the paper needs a major overhaul. 

The authors thank the reviewer for bringing this point of confusion to our attention. Please refer to the discussion section, lines 393-398 for an updated description, correlating concentration gradient with behavior. 

Reviewer #3: PLOS ONE 

Manuscript No. PONE-D-22-14066 

Authors: Siegel EL, et al. 

1. Many of the explanations and descriptions given in this manuscript were weak and appeared rather hurriedly done. The title of the manuscript claims that this work represents a new paradigm in evaluating spatial repellents to ward off tick parasitism. Such a paradigm will be welcomed by many. But if this work is to truly present a new way forward, then everything – from describing and illustrating the apparatus to providing methods of calculating and analyzing outcome variables – needs to be explained in a more clear and concise manner. Some of the ways this can be done are listed below. There were 6 behavioral outcomes (i.e., dependent variables) used to assess the spatial repellency of 4 different volatiles. Some of the variables (e.g., climbing success versus climbing height reduction) appeared to be highly interdependent. The manuscript would increase its usefulness to other vector biologists if authors could either rank the variables or make a recommendation, based on their experience, on which of the six might are the most useful and meaningful to employ when testing volatiles for spatial repellency of host-seeking ticks. 

The authors thank the reviewer for raising these valid points. The title of the manuscript has been revised to better reflect the goals, results, and conclusions of the study. Further, more detailed information regarding the definition, calculation, and interpretation of the parameters is throughout the paper. The authors have reduced the parameters from 6 to 4 to reduce the confusion and focus on those more important. Please refer to the discussion section, lines 377-388, 405-416 for more information. 

2. INTRODUCTION. 

Lines 102-103. This is the last sentence in the final paragraph of the Introduction where authors typically state the research objective of their study. But the sentence construction here is jagged. There are several dependent clauses all strung together at the end of an already long sentence. This tends to muddle the meaning. Suggest condensing or break it into two sentences. 

The authors appreciate the reviewer’s comment on the structure of the final paragraph of the introduction. Please refer to lines 105-110 for updated text. 

3. Description of the apparatus needs improvement. For example: 

- Line 126. Is a ‘chemical-emanating device’ the same as a ‘controlled release device’? If so, then please refer to the device using a single, consistent term. Otherwise, it is confusing. 

- Line 129. Before describing the behavioral test chamber, it is appropriate to provide a description of the controlled release device. 

- Figure 1. Please provide dimensions of the controlled release device. 

- Line 135. “placement of the active ingredient” – shouldn’t that be “placement of the CRD containing the AI being tested”? 

- Line 138. Please state the composition of the sticks. Wooden sticks? Plastic sticks? Paper sticks? 

- Figure 2 legend; Line 148. Change the word ‘device’ to ‘controlled release device [CRD]” so that it matches what is shown in the diagram. 

- Line 159. Again, is the term ‘emanating device’ the same thing as the ‘controlled release device’? Please use consistent terminology throughout. 

- Line 182 – Video Tracking. EthoVision is a software program. Please provide brief description of the hardware and conditions used – e.g., camera, level of magnification if any, etc. This can be provided as a Supplement section. 

The authors thank the reviewer for noting where the materials and methods should be revised to improve clarity. CRD is now used consistently throughout the paper to avoid confusion. Section 2.3.1 details the controlled release device pictured in fig 1. This section shows the dimensions of the device. The methods have also been revised to include the information of the climbing sticks, lines 149-150) and the video tracking hardware (lines 179-180). 

4. RESULTS – 6 outcome variables. WHICH IS THE MOST IMPORTANT ?? 

The authors thank the reviewer for the comment and acknowledge that the definition and interpretation of the parameters were unclear. The materials and methods have been updated to fix this. 6 parameters have been reduced to 4. The way that each parameter was defined, calculated, and interpreted has been expanded on. Further, these have been put in context with tick ecology. Please refer to the discussion section for expansion on this topic, lines 377-388, 405-416. 

5. Tables – keep the order of tick species consistent – e.g., Amblyomma – Dermacentor – Ixodes. Mixing the order all around from one table to the next adds unnecessary confusion for the reader. 

Figures. – similarly, keep the order of the test compounds consistent from table-to-table and figure-to-figure (e.g., Fig. 5 differs from Figs. 6-8). 

The authors thank the reviewer for raising this issue with table and figure presentation. Please figures 5-8, which have been updated to keep the species and chemicals consistent. 

OUTCOME VARIABLES 

6. Tick Detachment – Table 1 Authors should point out that none of the 4 AI’s significantly altered detachment for A. amblyomma. The significant effect of metofluthrin on tick detachment was due solely to the response of D. variabilis and I. scapularis ticks. Thus, the statement on line 416 is incorrect. With A. americanum and metofluthrin, detachment in the control group was 2/15. Detachment in the metofluthrin group was 3/15 (Table 1). No difference. 

The authors thank the reviewer for raising this issue with table 1’s presentation. Table 1 has been removed. Figure 6 now shows detachment information. Please refer to section 3.2.2, lines 318-328 for updated results. 

7. Mean displacement – Table 2 - Data were not normally distributed (lines 201-202). Therefore, values should be presented as geometric means and 95% CL, not arithmetic means + SD. 

The authors agree with the reviewer and appreciate the feedback. Mean displacement has been removed as a parameter from the results. Other analyses have been updated to present medians and interquartile ranges. 

8. Climbing height reduction – Fig. 5 – What do the numbers signify atop each histogram ? Percentages? They should also indicate the sample size. Line 192. Climbing height reduction needs to be more fully explained – e.g., how were heights of AI-exposed ticks ‘normalized’ to non-exposed control ticks? Was there some kind of mathematical formula used? If so, please provide. Climbing Success – Fig. 6 – what is the difference between Climbing Success and Climbing Height Reduction. This should be explained clearly because the pattern of histograms (Fig. 5 & Fig. 6) appear nearly identical for two of the three tick species. Are you showing the same thing or something different in Figs. 5 & 6. This should be made clear. 

The authors agree with the reviewers that there were clarity issues in defining the parameters. Climbing height reduction is no longer presented. Climbing success is now presented as climbing deterrence. Please refer to the materials and methods, section 2.5.2, lines 200-211, for further information on how this was defined. The results for this analysis are shown in section 3.2.3, lines 331-344. 

9. Much of the text in Figure legends – Fig. 6, 7, 8 – should be taken out of the figure legends and placed in the appropriate place within the text of Results section.FIGURES 3, 4, 7 & 8. The axes labels are too small and need to be enlarged to be legible.FIGURE 3a & 3b. No exaggeration, I had to use a magnifying glass to read the labels on the y-axes. The x-axis was not labelled at all. 

The authors thank the reviewer for raising clarity issues with these figures. The figures have been re-made to reflect changes to the figure legend content, font size, and presentation. 

10. Where is the CRD placement in relation to the configuration of this chart? Presumably at top left corner for larger box, but at top right for smaller box? This needs to be clarified. FIGURE 7. Likewise, the font chosen to express significance level is too small. 

The authors thank the reviewer for bringing our attention to this source of unclarity. Please refer to the figure legend, lines 263-271, for the change reflecting identification of the CRD in the figure. 

11. TABLE 3. Use a footnote or some other way to denote what the asterisks signify. Also, use a zero to the left of the decimal point, otherwise a reader might overlook the small little dots (decimal points) in a table containing 12 rows of data. 

The authors thank the reviewers for raising this concern and acknowledge the need for clarity. Footnotes have been included in figures 5-8 to include designated levels of significance. A zero to the left of decimals is now used throughout the manuscript. 

12. Line 204. States that all ticks were included. But lines 174-175 states that some ticks were excluded. Which is correct? 

The authors thank the reviewer for raising points that were unclear in the previous version. The manuscript has been updated to better address these comments. Please refer to the materials and methods, section 2.5.1 (lines 189-199), for an updated and clarified description on the activity assessment. Further, the results, section 2.5.1 (lines 190-199), and discussion, lines 404-429) have been revised to address these concerns. Immobility should not have been understated. The methods have been revised to clarify that ticks that remained immobile have been considered in an updated analysis, where they are only excluded to evaluate tick velocity while moving along the stick. 

13. Line 347 – What is meant by the phrase “build on the shortcomings”? Building anything new based on the defective ideas/practices of the past does not a seem like the best approach for progress. Is this really what the authors meant to say? 

The authors appreciate the reviewer’s comment and agree that this is not an appropriate statement. The discussion has been revised and this is no longer included. 

14. Line 408 – “pushing ticks to continue questing for a safer place”? Not sure what this means. Is there any evidence to support the notion that ticks are sentient enough to sense danger? Perhaps it is better to frame this in terms of a mechanistic response to volatiles 

The authors thank the reviewer for raising this valid point. The discussion has been revised to reflect changes, lines 389-416. 

15. Lines 462-464. The ideas contained in this sentence are not relevant to a discussion of spatial repellents distributed by controlled release devices. To avoid confusing the use of personal repellents with spatial repellents, this sentence should be removed. 

The authors appreciate the feedback from the reviewer and acknowledge that this statement is confusing and inappropriate in this context. The manuscript has been revised and this statement has been removed. 

16. MINOR GRAMMATICAL & MISCELLANEOUS 

Line 65. Add word “repellent” between the words ‘spatial’ & ‘compound’. 

Line 67. Change ‘are’ to ‘is’; or delete phrase “A new generation of …”. 

Lines 75 & 83. Change the word ‘arthropods’ to ‘insects’. Insects are the only arthropod capable of flight. Therefore flying arthropods are by default, flying insects. 

Line 109. :.. make test 30% test …” Remove the 1st test? 

Line 115. State the species of wild ticks that were collected in North Amherst, MA used to supplement the OSU colony ticks. 

Line 192. Something is missing in this sentence. 

Line 239. Species name should be written as lower case. Watch out for auto-correct. 

Lines 237 – 247. Authors switch from past tense to present tense, then back to past tense. Suggest sticking with just past tense in reporting results. 

There are 13 abbreviations used throughout this manuscript. Are they all necessary? If you only use a term once or twice (e.g., ULV) why is it necessary to mark it with an abbreviation? Likewise, it is never good to begin a sentence with an abbreviation. Just as the first word in a sentence should be capitalized, the subject of a sentence should be spelled out. 

The authors thank the reviewer for acknowledging these issues with grammar and presentation which warranted change. The compounds discussed are now described as spatial repellent compounds in text as opposed to spatial compounds. Grammatical issues with tense, abbreviation over/misuse, and tense has been addressed throughout the paper. Only ticks included from OSU colony are included in the present analysis. The methods, results, and conclusions have been updated to reflect this change.

---

## [Decision Letter · Decision Letter 1]

12 Oct 2022

PONE-D-22-14066R1Spatial repellents transfluthrin and metofluthrin affect the behavior of *Dermacentor variabilis*, *Amblyomma americanum*, and *Ixodes scapularis* in an *in vitro* vertical climb assayPLOS ONE

Dear Dr. Rich,

Thank you for submitting your manuscript to PLOS ONE. After careful consideration, we feel that it has merit but does not fully meet PLOS ONE’s publication criteria as it currently stands. Therefore, we invite you to submit a revised version of the manuscript that addresses the points raised during the review process.

 Please address the very minor comments from the second reviewer, who brought up some items for clarification, particularly with Materials and Methods as you are describing a novel assay. I apologize for the delay, the original slate of reviewers were not all available. I look forward to your revision and will process it promptly.

We look forward to receiving your revised manuscript.

Kind regards,

Catherine A. Brissette, Ph.D.

Academic Editor

PLOS ONE

Journal Requirements:

Reviewers' comments:

Reviewer's Responses to Questions

**Comments to the Author**

1. If the authors have adequately addressed your comments raised in a previous round of review and you feel that this manuscript is now acceptable for publication, you may indicate that here to bypass the “Comments to the Author” section, enter your conflict of interest statement in the “Confidential to Editor” section, and submit your "Accept" recommendation.

Reviewer #1: (No Response)

Reviewer #4: (No Response)

2. Is the manuscript technically sound, and do the data support the conclusions?

Reviewer #1: (No Response)

Reviewer #4: Yes

3. Has the statistical analysis been performed appropriately and rigorously? 

Reviewer #1: (No Response)

Reviewer #4: Yes

4. Have the authors made all data underlying the findings in their manuscript fully available?

Reviewer #1: (No Response)

Reviewer #4: Yes

5. Is the manuscript presented in an intelligible fashion and written in standard English?

Reviewer #1: (No Response)

Reviewer #4: Yes

6. Review Comments to the Author

Reviewer #1: (No Response)

Reviewer #4: (No Response)

7. PLOS authors have the option to publish the peer review history of their article (what does this mean?). If published, this will include your full peer review and any attached files.

Reviewer #1: No

Reviewer #4: No

---

## [Author Response · Author response to Decision Letter 1]

20 Oct 2022

General comments: Spatial repellents transfluthrin and metofluthrin affect the behavior of Dermacentor

Variabilis, Amblyomma americanum, and Ixodes scapularis in an

in vitro vertical climb assay

The authors thank the reviewer for the feedback. All individual points raised have been addressed as noted below: 

The manuscript is a well-detailed investigation of a relatively low-complexity test method and parameters for scoring when using semi-volatile active ingredients against ticks. I do sort of wonder at the labor-requirement in using videography for scoring, but for the most part I think this is a very accessible testing architecture for other research groups. I was not part of the first review cycle, so, as my first review of this work, I feel that there is a lot of written polish and good logic presented throughout. I have only minor suggestions to help strengthen the paper. Otherwise, I believe this is an excellent contribution to tick research and spatial repellent methodology.

Abstract:

My personal preference is to be a bit more specific in the abstract. These are minor suggestions:

1. Line 38 – What are these parameters?

The authors thank the reviewer for the feedback regarding the abstract structure. Please refer to lines 39-40 on the updated manuscript, where the parameters are now defined. 

2. Line 39-41 – To what extent? Can you give something a bit more concrete and quantitative, like a % of ticks that refused to cross a certain distance into the plume or something?

The authors thank the reviewer for this additional suggestion in the abstract and agree that this addition strengthens it. Please refer to lines 42-46, where the parameter which we feel is most important/representative of the study goals has been quantified to give the readers a better idea of the effect of spatial repellents in this study. 

Introduction:

3. Line 57 – Could you be more specific? Relay to what extent incidence has risen (such as a range of case increases or something). Dramatically isn’t very easy to assign a value to.

The authors thank the reviewer for raising this point. We agree that adding this information is useful to the readers. Please refer to lines 56-59 in the updated introduction, where this is now revised.

4. Line 64 – I would argue that personal protection is the only true pillar of tick bite prevention, as there are no meaningful trapping methods (tick drags are very labor intensive and inconsistent) and chemical control is laughable (routinely dousing leaf litter with broad spectrum insecticides is neither targeted nor very effective; only desperate).

The authors thank the reviewer for this comment. The language used in line 65 has been revised to better represent the role that personal protection plays in tick bite protection. 

The rest of the introduction is very comprehensive and well written. I like the articulated argument made about redefining repellency standards for ticks and the logic given for why that matters.

Materials and Methods:

5. Line 114-115 – Please indicate the purity of your starting material (ex: 94% technical grade). Also, generic supplier is not very informative; Sumitomo still controls the majority of product flow for metofluthrin. There are varying isomeric purities that matter for all pyrethroids, including the penchant for R-isomerization that even the polyfluorinated structures exhibit. I think you should be more specific about the supplier. If it is in-house or proprietary, that is important to be aware of as well.

The authors appreciate the feedback regarding the materials section. The sourcing, technical grade, and isomeric composition of the active ingredients has been revised where needed. This can be found on lines 114-117. 

6. Line 126 – 127 – I assume you were maintaining a chill coma? You should specify if you were clearly inducing chill coma because there are also more efforts recently that study the cold-hardening processes in ticks, which lowers the chill coma temp at the expense of resilience.

The authors thank the reviewer for raising this point of clarity in the materials and methods. The storage and handling information has been revised to clarify that a chilled coma was maintained until the ticks were used. This can be found on line 128. 

7. Line 149 – So this was not a ventilated box? Please clearly indicate if this was a static air apparatus, particular since there is some division in literature between studies doing static air olfactometry versus filamentous/laminarized plume-airflow systems.

The authors thank the reviewer for raising this additional point of clarity in the materials and methods. Please refer to lines 151-152, where it has been clarified that ventilation occurs between the outside air and the air in the chamber. 

8. Line 165 – Please reiterate here your dilution to 30%; it’s easy to miss your one mention of it on Line 116.

The authors agree with the reviewer that this should be restated. Please find this done on like 168. 

Results/Discussion:

The authors did an excellent job of bridging their investigations to the intended root of application in most ways (good bioassays, good exposure paradigms, good bridging with IPM practices). 

I don’t have much to argue with in the bulk of their points as they are well formulated thoughts with a plethora of literature. I will point out just a couple “watch outs.”

9. Line 502 – 505 – I would counter that the main limiter to spatial repellents against ticks is the poor integrity of the gradient. Either because of increased vapor pressure resulting from humidity or losing the envelope of protection to air movement/actively walking out of the vapor. Therefore, I personally think there is much greater risk the CRD will not protect a large enough area to prevent bites. You hint at this to some degree with your data showing a very visible loss in magnitude of concentration within only 20 cm.

The authors thank the reviewer for the feedback in the discussion section. Lines 501-504 have been revised to include changes in vapor pressure and stress that the integrity of the gradient barrier, which serves as the basis of protection with CRDs, may be affected with certain external factors.

---

## [Editor Report · Decision Letter 2]

25 Oct 2022

Spatial repellents transfluthrin and metofluthrin affect the behavior of *Dermacentor variabilis*, *Amblyomma americanum*, and *Ixodes scapularis* in an *in vitro* vertical climb assay

PONE-D-22-14066R2

Dear Dr. Rich,

We’re pleased to inform you that your manuscript has been judged scientifically suitable for publication and will be formally accepted for publication once it meets all outstanding technical requirements.

Kind regards,

Catherine A. Brissette, Ph.D.

Academic Editor

PLOS ONE